

# Accelerated impact of airborne glaciogenic seeding of stratiform clouds by turbulence

Meilian Chen[1], Xiaoqin Jing[1], Jiaojiao Li[1], Jing Yang[1,2,*], Xiaobo Dong[3], Bart Geerts[4], Yan Yin[1], Baojun Chen[2], Lulin Xue[5], Mengyu Huang[6], Ping Tian[6], and Shaofeng Hua[2]

[1]Collaborative Innovation Center on Forecast and Evaluation of Meteorological Disasters (CIC-FEMD)/China Meteorological Administration Aerosol-Cloud and Precipitation Key Laboratory, Nanjing University of Information Science & Technology, Nanjing, 210044, China.
[2]CMA Cloud-Precipitation Physics and Weather Modification Key Laboratory (CPML), Beijing, 100081, China.
[3]Hebei Provincial Weather Modification Center, Shijiazhuang, 050021, China.
[4]Department of Atmospheric Science, University of Wyoming, Laramie, WY, 82071, USA.
[5]National Center for Atmospheric Research, Boulder, CO, 80305, USA.
[6]Beijing Weather Modification Center, Beijing, 100089, China.

*Correspondence to*: Jing Yang (jing.yang@nuist.edu.cn)

**Abstract.** Several recent studies have reported complete cloud glaciation induced by airborne-based glaciogenic cloud seeding over plains. Since turbulence is an important factor controlling mixed-phase clouds, including ice initiation, snow growth, and cloud longevity, it is hypothesized that turbulence may have an impact on the seeding effect. To understand the role of turbulence in seeded clouds, idealized WRF large eddy simulations (LESs) over flat terrain are conducted for a shallow stratiform cloud in which complete glaciation was observed. The results show that the model can reasonably capture the magnitude and spatial distributions of radar echoes in seeded areas. Sensitivity tests suggest that, for this case, stronger turbulence enhanced particle dispersion, the nucleation of silver iodide (AgI) particles, and the growth of ice crystals, which accelerated cloud glaciation, even though the condensation of droplets was also enhanced. The faster cloud glaciation intensified precipitation within a short time after seeding, while the liquid water was quickly consumed, leading to a decrease in precipitation rate in the further downwind areas. Such a transition from positive to negative seeding effect is more pronounced for seeding with a higher AgI release rate. This study provides strong evidence that turbulence plays a vital role in the physical chain of events associated with cloud seeding.

## 1 Introduction

For more than half a century, clouds have been seeded operationally in many arid and semi-arid regions to enhance precipitation artificially (Rauber et al., 2019; Wang et al., 2021; Geerts and Rauber, 2022). Silver iodide (AgI), which has a similar crystal structure to that of ice, is the most widely used glaciogenic seeding material because it can act as ice nucleating particles (INPs) at temperatures higher than most aerosols (DeMott 1997). It has been demonstrated that AgI seeding can enhance precipitation under suitable conditions based on recent field experiments such as the 2017 Seeded and Natural Orographic Wintertime Clouds: The Idaho Experiment (SNOWIE; French et al., 2018; Tessendorf et al., 2019; Friedrich et al.,





2020). However, in most cases, the seeding impact usually cannot be readily identified as the radar seeding signatures are often obscured by the large variability of natural precipitation (Geerts and Rauber, 2022; Zaremba et al., 2024). For a radar signature

to be attributed unambiguously to seeding, the seeding-induced cloud phase relaxation time should be short compared to the characteristic time of natural dynamical and microphysical processes such as turbulent mixing, and cloud glaciation should be traceable to seeding release (French et al., 2018).

A decrease in cloud top or complete cloud clearing following aerial seeding is often regarded as a sign of efficient seeding

(though it does not indicate enhancement in surface precipitation) (Mason, 1957; Wallace and Hobbs, 2006; Rogers and Yau, 1989). It is a result of complete cloud glaciation, which consumes liquid water faster than mixing or liquid water formation by dynamic forcing. This phenomenon has not been observed when seeding is conducted over mountains such as that in SNOWIE, because the orographic lifting can continuously provide liquid water through condensation. Also, a dynamic (buoyant) response to the latent heat released by cloud glaciation can raise the cloud top (Bruintjes, 1999). However, decrease in cloud

top has also been reported in several studies in which seeding experiments were conducted over plain regions (e.g., Yu et al., 2007; Li et al., 2021; Wang et al., 2021). Although the complete cloud glaciation is helpful in identifying seeding signatures, it indicates that there may be insufficient liquid water and thus precipitation suppression downwind of the target areas, leading to the so-called "robbing Peter to pay Paul" phenomenon (Long, 2001; DeFelice et al., 2014). Some studies argue that the positive seeding effect may extend to 50-200 km downwind of the target area (e.g., Solak et al. 2003; Griffith et al. 2009;

DeFelice et al. 2014; Mazzetti et al., 2023). If their results are valid, it implies a continuous liquid water supply along the seeding impacted areas. Using X-band radar data collected in the 2012 AgI Seeding Cloud Impact Investigation (ASCII) experiment conducted over mountains in Wyoming (Geerts et al., 2013), Jing et al. (2016) found the enhancement of precipitation by seeding can extend to 50 km (limited by the radar detection range). Their study, as well as Xue et al., (2014; 2016), highlighted that mechanisms such as hydraulic jump, lee convection or turbulence in the lee of the target mountain are

vital in the vertical dispersion of AgI particles and the generation of supercooled liquid water over the downwind mountain. Beyond-target (or "extra-area") positive seeding impacts have been documented also in SNOWIE cases (e.g., Fig. 15 in Xue et al., 2022).

In mixed-phase clouds over flat land (plains), turbulence is regarded as the most important mechanism for producing

supercooled liquid water and maintaining the mixed-phase clouds (Morrison et al., 2012; Korolev and Mazin, 2003). Without turbulence, a mixed-phase cloud can be completely glaciated in a few hours or less due to the Wegener–Bergeron–Findeisen (WBF) diffusional growth process, depending on the ice particle concentration (Rangno and Hobbs, 2001; Morrison et al., 2012). Korolev and Mazin (2003) proposed a formula for the minimum vertical velocity that is required to trigger the condensation of liquid water and the simultaneous growth of droplets and ice crystals. Korolev and Field (2008) point out that

turbulent fluctuations may not repeatedly produce a mixed-phase cloud as harmonic oscillations, but may maintain a long-lived mixed-phase environment. Hill (2014) confirmed the validity of the theoretical framework of Korolev and Mazin (2003)



in 3D large eddy simulations (LES), which further demonstrated a positive correlation between turbulence and liquid water content (LWC). Turbulence not only affects the formation of liquid water but also influences the growth of ice and snow particles in clouds (Chu et al., 2018). Turbulence can promote net ice growth and precipitation through alternating up- and down-drafts, while it can suppress this through cloud top entrainment of dry air (Chu et al., 2018). On the contrary, turbulence may result in pure liquid and ice clusters, which shrink the contact volume between ice and liquid water, thus the ice growth rate declines (Tan et al., 2016; Deng et al., 2024). Recently, based on LES, Yang et al. (2024a) showed that mixed-phase clouds can be long-lived when there is a balance among liquid water generation, ice growth, and turbulent mixing.

According to the studies shown above, it is evident that turbulence is helpful to continuously provide liquid water in mixed-phase clouds, thus a scientific question is raised: Is seeding in stronger turbulence helpful to avoid the "robbing Peter to pay Paul" effect and extend the positive seeding effect downwind of the target areas? If not, how does turbulence affect the seeding effect? To address this question, this paper investigates the physical responses of cloud microphysics and precipitation to turbulence using LES. The simulations are done over flat land, in order to focus on the effect of turbulence, but the results are relevant to mountains, where most operational cloud seeding is conducted. A case with complete glaciation observed in the seeding plume is selected for the simulation, and sensitivity tests by altering the turbulent strength are conducted. The results will deepen our understanding of the impacts of turbulence on the glaciogenic cloud seeding effect, and further explain the competition among liquid water condensation and cloud glaciation in mixed-phase environments.

The rest of the paper is organized as follows. Section 2 describes the case and the model setup. In section 3, the model results are evaluated using radar and satellite measurements, and the impacts of turbulence on the seeding effect are analyzed. A discussion and the main findings are presented in Sections 4 and 5, respectively.

## 2 Case description and model setup

### 2.1 Case description

On 20 Jan 2022, an airborne glaciogenic cloud seeding experiment was conducted in Hebei Province over the North China Plain. A persistent supercooled cloud was documented in the flight area before it was seeded. No surface precipitation was observed, and no radar echo was detected by the ground-based S-band radar, suggesting minor or no natural ice formation in the cloud. Weak low-level baroclinicity was present, with colder air to the north, and weak southerly flow provided sufficient water vapor, resulting in a high ambient relative humidity (Fig. 1a). At 500 hPa, dry westerly flow around a weak ridge dominated in the flight area (Fig. 1b). The environment was synoptically quiescent and stably stratified. A stratiform cloud deck was present (Fig. 1c). This cloud was decoupled from the surface, only ~500 m deep, non-precipitating (Fig. 2), and with a cloud top temperature of about -16 °C.



Seeding was conducted at the cloud top at about 8:00 UTC, as shown by the red lines in Fig. 1c. The true air speed of the
aircraft was approximately 100 m s$^{-1}$, and the release rate of AgI particles was $10^{14}$ s$^{-1}$ as estimated based on the mass burned
per second using burn-in-place pyrotechnic flares. The aircraft flew one and half north-south oriented legs while seeding (Fig.
1c). Any seeding signatures would have to reveal the same spatial pattern in the cloud/precipitation field (advected downwind),
as illustrated vividly in photographs of the first airborne cloud seeding experiments (Schaefer, 1949) and reproduced in many
meteorology textbooks (e.g., Rogers and Yau, 1989; Lutgens et al., 2006). In areas unaffected by seeding, the cloud top was
fairly flat as seen from the visible images detected by the FY4A satellite at 8:15 UTC (Fig. 1c), and the brightness temperature
at 12 μm varied between -12 °C and -14 °C (Fig. 1d). Clear seeding signatures were detected downwind of the seeding line on
both the visible and infrared images. The IR brightness temperature increased by about 2 °C along the advected flight track,
indicating that the clouds become thinner and the cloud tops descend due to the consumption of supercooled liquid water by
growth of ice crystals. The more resolved visible image reveals a reduction in cloud brightness along the same track, indicating
a reduction in droplet concentration, and an enhancement in reflected solar radiation along the (north)east flank of the cloud
top depression (At 8:00 UTC the sun is in the southwest in the flight area). The displacement of the satellite signature relative
to the flight track is consistent with the wind speed and direction at the cloud top level.

The seeding signatures were also detected by the ground-based S-band radar (Fig. 2), which is located north of the flight area.
The scan at 1.5° best captured the enhanced reflectivity since the cloud depth was only about 500 m. As seen in Fig. 2a, the
radar echo appeared about ten minutes after seeding. This is consistent with radar data collected during airborne AgI seeding
of a shallow stratus cloud deck under quiescent synoptic conditions in Switzerland (Henneberger et al., 2023) and continuously
strengthened till 8:42 UTC. The reflectivity varied between -10 dBZ and 10 dBZ, and it gradually weakened after 08:48 UTC.
Since the cloud was mostly liquid and no surface precipitation was observed in areas unaffected by seeding, we are not able
to investigate the downwind effect of precipitation using observations. However, we may conclude that the liquid water that
is available for precipitation downwind of the target area has been reduced by the seeding operation. The downwind effect will
be discussed using simulations in Section 3.



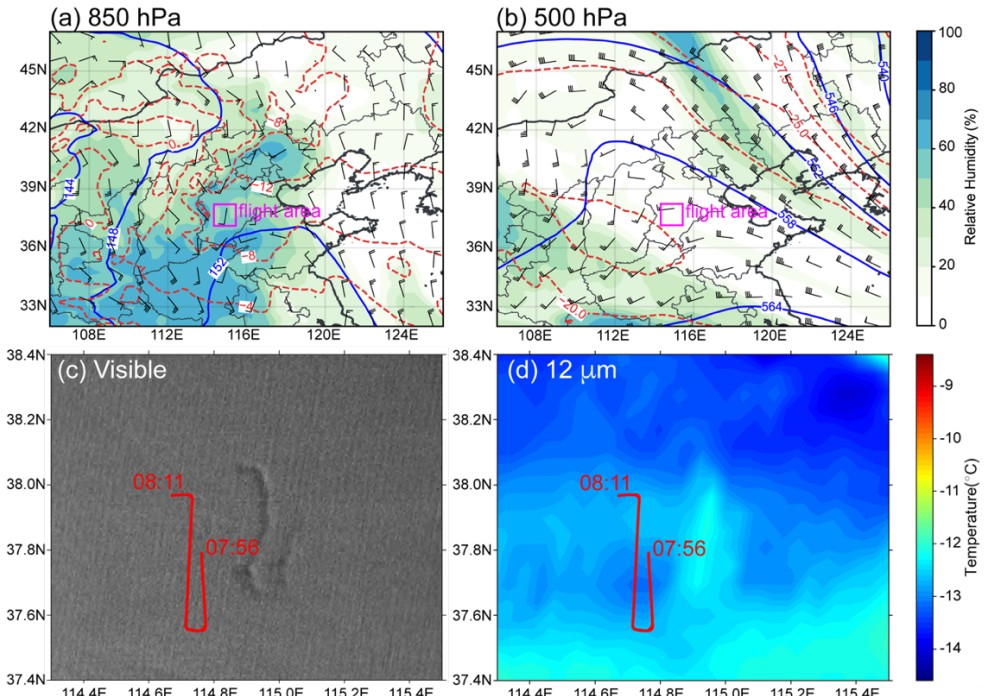

**Figure 1. (a, b) Synoptic conditions at 850 hPa and 500 hPa in North China at 06:00 UTC on 20 Jan 2022 obtained from**
**ERA5 reanalysis data, including the geopotential height (dam, blue contours), isotherms (°C, red contours), wind barbs,**
**and relative humidity (shaded). (c) Visible image and (d) brightness temperature at 12 μm obtained from FY4A satellite**
**at 08:45 UTC. The red lines indicate the seeding trajectory.**

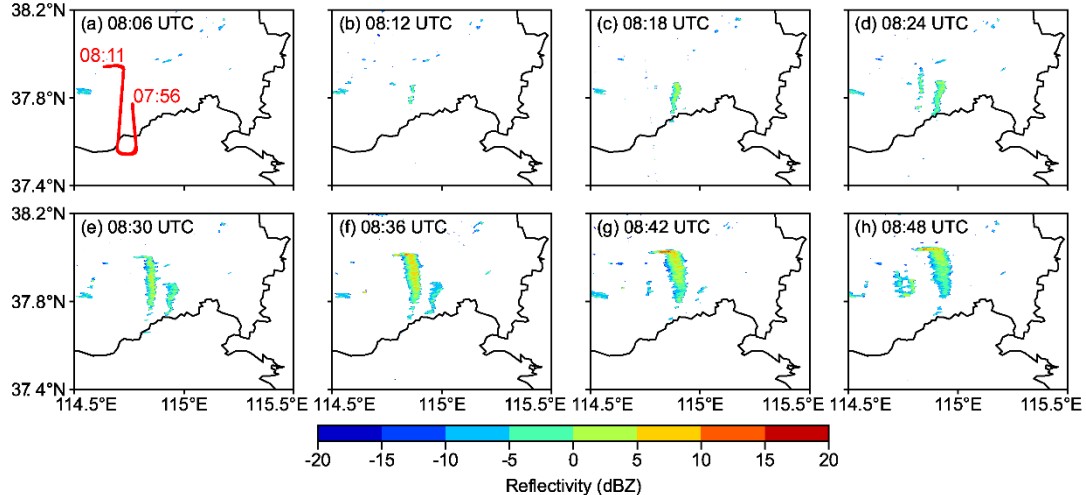

**Figure 2. The radar reflectivity at 1.5° elevation from 08:06 UTC to 08:48 UTC measured by a ground-based S-band**
**radar located north of the flight area.**



## 2.2 Model setup

The LES mode in the Weather Research and Forecasting model is used to conduct idealized numerical simulations for the case
described above. To reproduce the entire seeding trajectory, we use a domain size of 80 km × 80 km × 3 km with periodic
lateral boundaries. The surface is assumed to be flat. The model has a horizontal resolution of 100 m and 90 levels in the
vertical direction. The seeding trajectory is the same as that performed in the field experiment, in which the aircraft flew at a
speed of 100 m s$^{-1}$ along the red line in Fig. 1c, and released the AgI particles near the cloud top. The AgI particles are assumed
homogeneously mixed in a grid box as soon as it is released from the aircraft. The physics schemes used in the simulation
include the fast spectral bin microphysics scheme (Khain et al., 2004), the Revised MM5 surface layer scheme (Jiménez et al.,
2012), the Noah Land Surface Model (Tewari et al., 2004), and the Rapid Radiative Transfer Model (Mlawer et al., 1997).
Cumulus and boundary layer parameterization are turned off in the LES. The cloud condensation nuclei (CCN) concentration
is expressed by $N_{CCN} = N_0 S_w{}^k$, where, $N_0$ refers to the CCN concentration at a supersaturation level of 1%, $S_w$ represents the
supersaturation with respect to water (%), $k$ is the slope of the CCN size distribution. For the continental area of China, $N_0$ =
4,000 cm$^{-3}$ and $k = 0.9$ at surface are assumed.

The parameterization of AgI nucleation implemented in the fast spectral bin microphysics scheme was developed by Xue et al
(2013), including four nucleation modes: deposition nucleation, condensation nucleation, contact freezing, and immersion
freezing. The fraction of AgI aerosols that can nucleate is confined to a specific range of temperature and supersaturation ratio,
and the sum of the four nucleation modes fractions cannot exceed one. The contact and immersion freezing modes require
distinct consideration of the proportion of AgI particles removed by droplets and the other nonactivated fractions immersed in
the droplets. Droplets collect AgI particles through Brownian diffusion, turbulent diffusion, and phoretic effects (Xue et al.,
2013). The majority of the AgI particles remain in the droplets after being removed from the air, while the remainder are
converted into AgI-containing hydrometeors via contact and immersion freezing. The activation process of AgI particles acting
as CCN is not considered in the model.

A single sounding is used to drive the model, i.e., the LES contains no horizontal heterogeneity initially, and large-scale
synoptic conditions do not evolve during the model time. Thermodynamic and wind profiles are shown in Figure 3. The
atmosphere is saturated at the altitude of 1372–1893 m. Above 1893 m, a strong inversion layer was present. We implemented
an initial liquid water mixing ratio profile which increases from 0 to 0.2 g kg$^{-1}$ from cloud base to 1893 m, and decreases
rapidly to 0 at cloud top (grey shaded area in Fig. 3b). The original wind speed is weak (solid lines in Fig. 3c), which results
in a weak turbulent environment. To investigate the effect of different turbulence intensities on the seeding effect, we follow
the method in Hill et al. (2014), which shows that enhancement of vertical wind shear in LES can intensify the modelled
turbulence. The wind shear between 1519 m and 1733 m height is enhanced by five times (dashed lines in Fig. 3c), this causes





a decrease in Richardson number in this layer from 16.81 to 0.67.

Figures 4a and b show the average turbulence intensities and turbulent kinetic energy (TKE) from the simulations with default and enhanced wind shears. It is seen that the turbulence intensity and TKE increase in the first hour, and there is no clear difference between the two simulations at this stage.. Therefore, a spin-up time of at least 1 hour is needed. The differences in

the turbulence intensity between the two simulations can be clearly seen after 1 hour, with the maximum difference around 3:00 Model Time (MT). The cross-sections of vertical velocity (Fig. 4c and d) prove that the updrafts and downdrafts are enhanced due to the stronger wind shear, and it is expected that the enhanced turbulence will influence the droplet activation, ice nucleation, and growth of hydrometeors.

In addition, to test whether turbulence plays the same role for varying AgI concentrations, we enhanced the AgI releasing rate by 10 times (i.e., $10^{15}$ s$^{-1}$) in both experiments with default and enhanced shears. The abbreviations for different experiments are listed in Table 1. To investigate the seeding effect, we will compare the cloud microphysics and precipitation in SEED and NOSEED areas, which are defined as the areas inside and outside the seeding plumes at each moment. Thus, the SEED and NOSEED areas moved with time along the direction of the prevailing wind. Since the observed cloud was mostly liquid before

seeding, we turned off natural ice nucleation when validating the model results (Section 3.1). However, to better understand the differences in ice generation and growth in SEED and NOSEED areas, we show analyses from experiments with natural ice nucleation turned on after the model validation section. The natural ice nucleation and the seeding of all experiments in Table 1 start 2:00 MT.

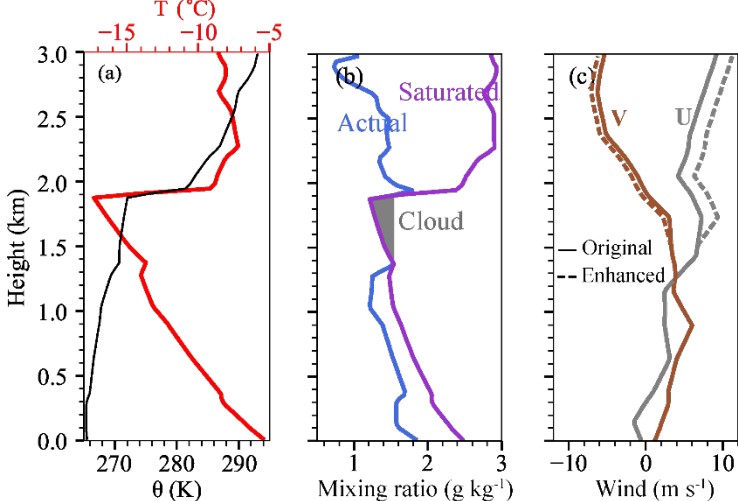

**Figure 3. The initial vertical profiles of (a) temperature and potential temperature, (b) actual vapor mixing ratio and saturation vapor mixing ratio relative to water, and (c) original and enhanced U and V components. The grey shaded area in (b) indicates the initial liquid water mixing ratio.**





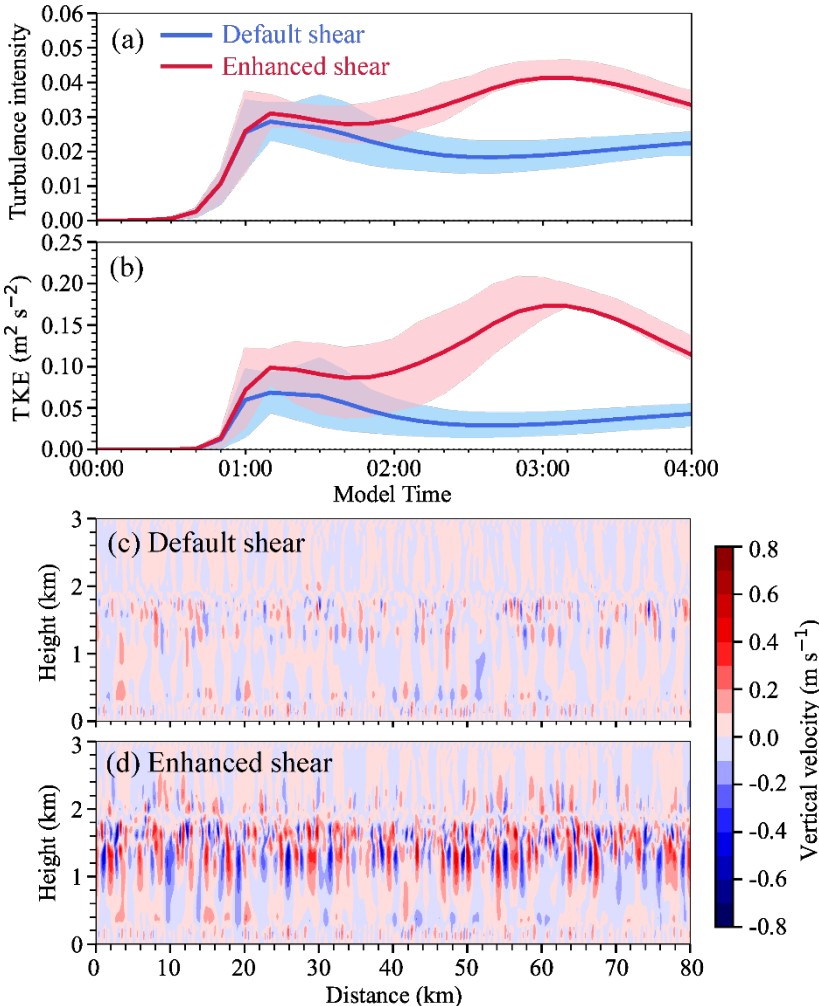

**Figure 4. Temporal variations of turbulence intensity (a) and TKE (b) from the simulations with default and enhanced wind shear. The upper and lower boundaries of the shaded areas respectively indicate the 75th and 25th percentiles, and the solid lines indicate the mean values. (c) and (d) are the cross-sections of vertical velocity at 03:00 Model Time from the simulations with default and enhanced wind shear, respectively.**

**Table 1. Design of numerical experiments.**

| Experiments | Enhanced wind shear | Enhanced AgI concentration |
|---|---|---|
| Control | No | No |
| EnWS | Yes | No |
| EnAgI | No | Yes |
| EnAgI/EnWS | Yes | Yes |





## 3 Results

### 3.1 Model evaluation

The modelled reflectivity from the Control simulation without natural ice nucleation is shown in Fig. 5. Since we use idealized LES, it is within the expectation that there are inevitably some differences between the modelled and observed results.

Therefore, we do not directly compare the model with observation here, we focus on evaluating the magnitude and temporal variation of radar reflectivity. It is seen from Fig. 5 that the spatial distribution of enhanced radar reflectivity is controlled by the seeding trajectory and moved northeasterly with time. With higher resolution, the model can produce finer structures of reflectivity distribution than the observation. The magnitude of modelled reflectivity varies between -12 dBZ and 12 dBZ, generally consistent with the observation shown in Fig. 2, but it seems that the horizontal dispersion of the seeding plume was

weaker in the modeled than the observed. The seeding signatures kept strengthening for about 1.5 hours before turning weaker, which is slightly longer than observed (Fig. 2), suggesting that in the actual cloud, the cloud glaciation is faster than in the modeled.

Due to the consumption of the supercooled liquid water, the cloud top height decreased, (identified using a threshold of total

water content greater than 0.001 g kg$^{-1}$), and in most of the SEED areas, the cloud top temperature increased by about 2 °C after 2 hours (Fig. 6), which is generally consistent with the observations. In the NOSEED area, the cloud top persisted at about 1.9 km (Fig. 5e-h). In the observation, cloud thinning was clearly seen almost in the entire seeding plume 15 minutes after seeding, while in the model, the increase in cloud top temperature was seen only in a small fraction of the SEED area within 1 hour after the seeding was performed. This difference again suggests that cloud glaciation was faster in the actual

cloud than in the model.

Such a difference could be due to multiple reasons. From the perspective of dynamics, large-scale forcing is not considered in the model, and we use a single-sounding measurement to drive the simulation, while in the real cloud, the wind field and cloud top stratification may change with time. From the perspective of microphysics, the study lacks measurements of CCN

concentration, so the droplet size and concentration may have uncertainties. In addition, the ice growth rate in the model may be underestimated because the crystal shape is not well considered in the model. A recent study by Yang et al. (2024b) showed that the ice growth rate would be higher at -15 °C if assuming a plate-like shape rather than a spherical particle. Regardless of the uncertainties, the model can reasonably reproduce the magnitude and spatial distribution of radar reflectivity, as well as the increase in cloud top temperature, providing confidence for us to investigate the impacts of turbulence on the glaciogenic

seeding effect using the model simulations.





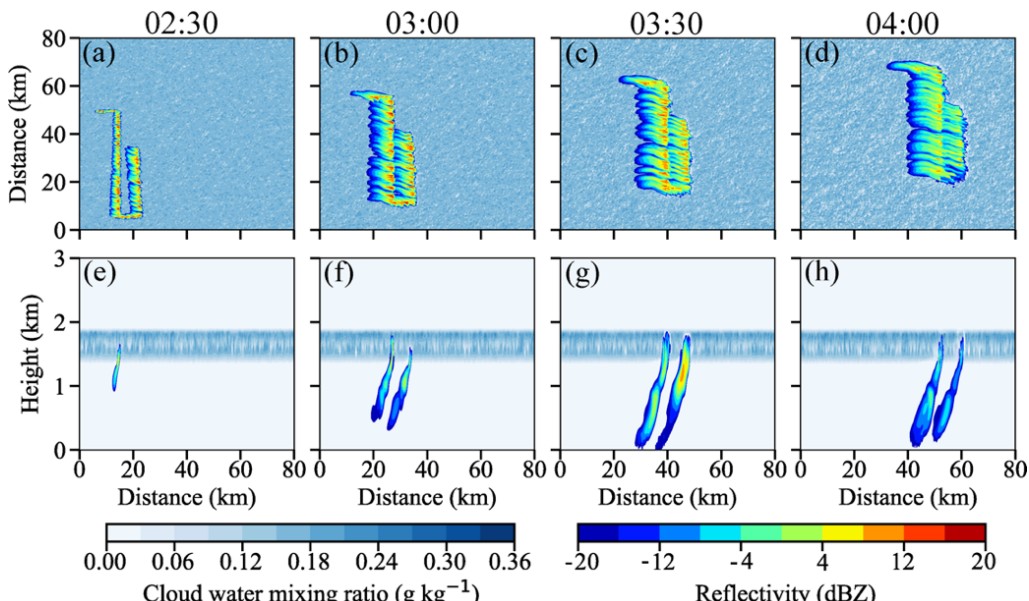

**Figure 5. (a-d) Maps of composite reflectivity and cloud water mixing ratio with natural ice nucleation turned off at seeding height from 02:30 to 04:00 Model Time. (e-h) Cross-sections of reflectivity and cloud water mixing ratios along y = 40 km from 02:30 to 04:00 Model Time.**


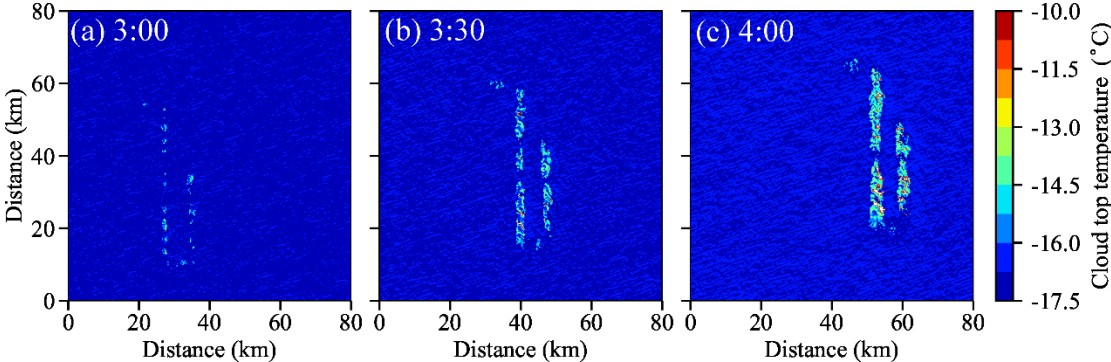

**Figure 6. Maps of cloud top temperature from 02:30 to 04:00 Model Time.**

## 3.2 Cloud microphysics

In this section, we investigate the changes in cloud microphysics after seeding was performed in the four different numerical experiments, in which natural ice nucleation is allowed. Figure 7 shows the cross-sections of radar reflectivity. It is seen that cloud seeding can induce enhancement in radar reflectivity. At 2:30 MT, the maximum reflectivity at a lower AgI releasing rate was about 15 dBZ (Fig. 7a and e), which is slightly larger than in the simulation without natural ice (Fig. 5). With a larger



AgI seeding rate, the maximum radar reflectivity reached 22 dBZ at 2:30 MT, when there was still sufficient liquid water to
support the ice growth (Fig. 7i and m). The stronger turbulence enhanced the horizontal dispersion of the AgI particles, so the
scales of seeding plumes were larger in the EnWS and EnWS/EnAgI experiments than the other two. In NOSEED areas, the
radar reflectivity was higher in the experiments with stronger turbulence, suggesting higher ice concentrations and larger ice
particles. Later, the radar reflectivity in SEED areas became larger in the EnWS and EnWS/EnAgI experiments than the other
two at 3:00 MT. However, the positive seeding effects attenuated more rapidly in the experiments with stronger turbulence
(Fig. 7g and o). At 4:00 MT, the EnWS and EnWS/EnAgI experiments obtained a negative seeding effect, and the liquid layer
(magenta lines) disappeared. While in the Control and EnAgI experiments, enhancement of radar reflectivity in SEED areas
can last for a longer time (Fig. 7d and l).

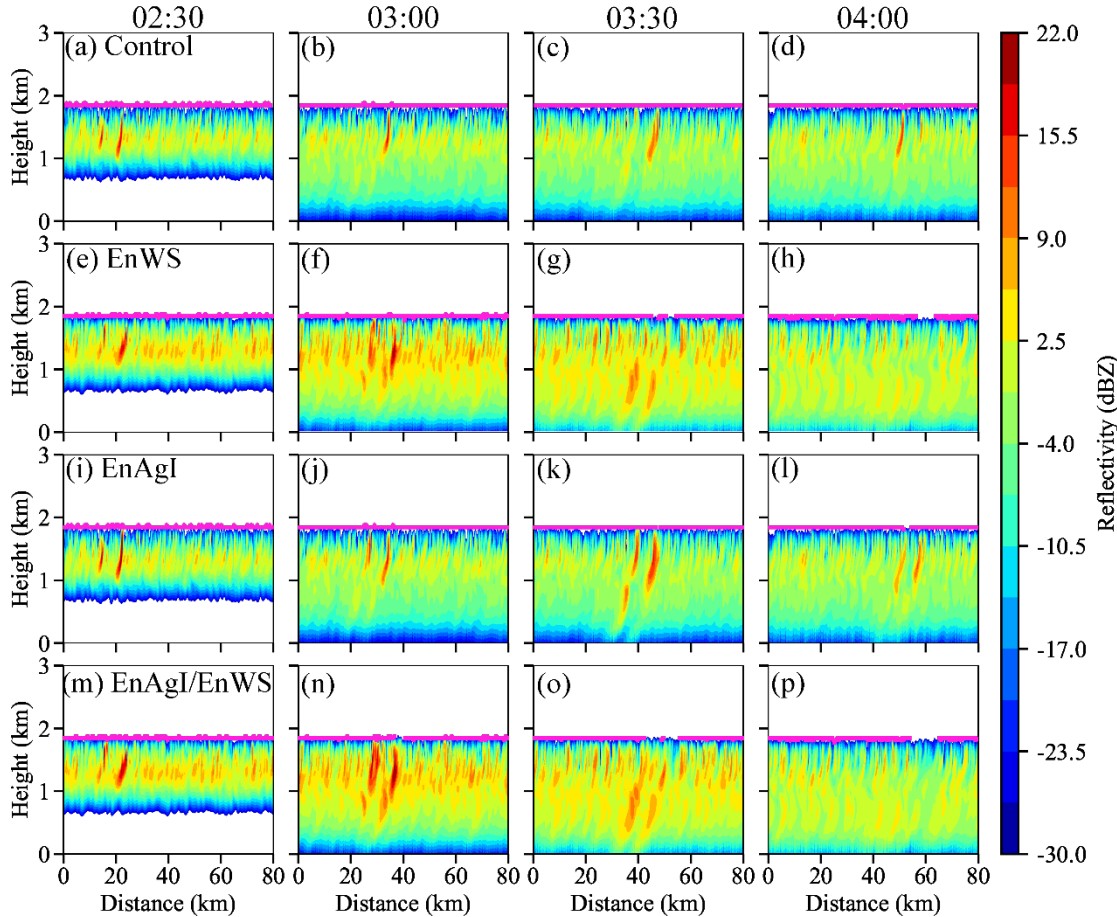

**Figure 7. East-west cross-sections of reflectivity from (a-d) Control, (e-h) EnWS, (i-l) EnAgI, and (m-p) EnAgI/EnWS**
**from 02:30 to 04:00 Model Time. The magenta lines indicate the top of the liquid layer. Natural ice nucleation is allowed**
**in the simulations. The cross-section are selected at y = 20 km, y = 30 km, y = 40 km, and y = 50 km at 2:30, 3:00, 3:30,**
**and 4:00 MT, respectively.**



The positive impact of turbulence on ice nucleation is also evident in the time-height diagrams of averaged ice concentration

and IWC (Figure 8). It can be seen that with stronger turbulence, the cloud obtained a higher ice concentration in SEED (green-shaded) areas soon after seeding (Fig. 8a, d, g, j), with the maximum value reached 7 L$^{-1}$ in the EnAgI/EnWS experiment, which is two times higher than that in the EnAgI experiment. However, in the SEED areas, the ice concentrations in the EnWS and EnAgI/EnWS experiments decreased rapidly with time, and became similar to that in the Control and EnAgI experiments after 3:20 MT. At the same time, stronger turbulence also promotes the nucleation of natural ice crystals in NOSEED areas

(black contours). The modelled natural ice concentrations in the experiments with enhanced turbulence have similar temporal variations as the turbulence intensity (Fig. 4a), and the maximum value exceeding 1 L$^{-1}$, is found between 2:50 and 3:20 MT. In the experiments with default turbulence intensity, the ice concentration was lower than 0.5 L$^{-1}$ and changed little with time after 2:20 MT. The higher concentrations of ice crystals tend to consume liquid water more rapidly as they grow, so it is seen that the decrease in LWC in the SEED area is the fastest in the EnAgI/EnWS experiment (Fig. 8l), resulting in a higher IWC

(up to 0.06 g kg$^{-1}$) before 2:50 MT. Although enhanced turbulence produced more supercooled liquid water, it is evident that the cloud glaciation rate was faster after seeding and the clouds rapidly became thinner. SEED areas completely glaciated when liquid water remained in NOSEED areas. This process involves competition among the liquid water generation, turbulent mixing, and cloud glaciation, which is analyzed in detail below.

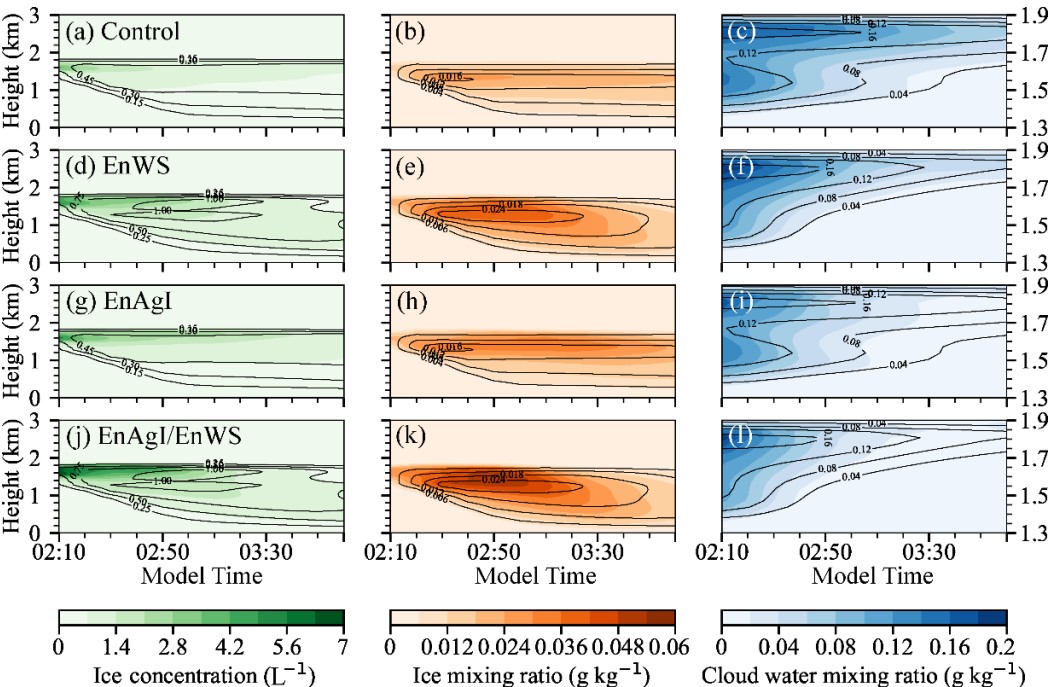

**Figure 8. Time-height diagram of ice concentration (left panels), ice mixing ratio (middle panels), and cloud water mixing ratio (right panels) from the (a-c) Control, (d-f) EnWS, (g-i) EnAgI and (j-l) EnWS/EnAgI experiments. The color-shading applies to the SEED areas, and the black contours are for the NOSEED areas.**



To better understand the ice growth and liquid water consumption at different turbulence intensities, the depositional growth
rate of ice crystals and condensation rate of droplets in SEED areas are plotted in Fig. 9. The ice growth is dominated by the
WBF process rather than riming and aggregation in this case (not shown). This is consistent with dual-polarization radar
measurements for seeded wintertime stratiform clouds (Jing et al., 2015; Jing and Geerts, 2015). It can be seen that turbulence
contributed significantly to the growth of ice crystals (Figs. 9a and b), and overall, the experiments with enhanced turbulence
had a higher deposition rate before 3:00 MT, and became lower than that in clouds with weaker turbulence due to the
insufficient liquid water supply. The experiment with higher AgI concentration produced greater changes in mass because of
more ice crystals. Figures 9c and 9d show the condensation rate of liquid water. In weak turbulence with relatively low AgI
concentration, the condensation rate varied mostly between 0 and $-2 \times 10^{-5}$ g kg$^{-1}$ s$^{-1}$, suggesting a relatively weak water
consumption (Fig. 9c). While in a stronger turbulent cloud, the condensation rate was greater between 2:00 and 3:00 MT,
indicating the generation of liquid water was significantly slower than its consumption by ice growth. With more AgI particles
seeded, the condensation rate shifted to negative in both simulations, indicating a faster cloud glaciation (Fig. 9d). At the cloud
tops, turbulence can promote the evaporation of liquid water due to the entrainment of dry air and detrainment of vapor, so
cloud glaciation was faster near cloud tops, even though entrainment of dry air may also suppress ice growth to some extent
(Chu et al., 2018). Significant entrainment is unlikely in this case, because of the very strong inversion just above cloud top
(Fig. 3).


In short, based on the analyses of Figs. 7-9, it is seen that stronger turbulence enhances the ice nucleation and ice growth in
the cloud. Even though stronger updrafts can provide more liquid water, the cloud in the SEED areas glaciated more rapidly
because the water consumption is faster than the water supply, and the turbulence is not able to maintain the cloud in mixed-
phase.




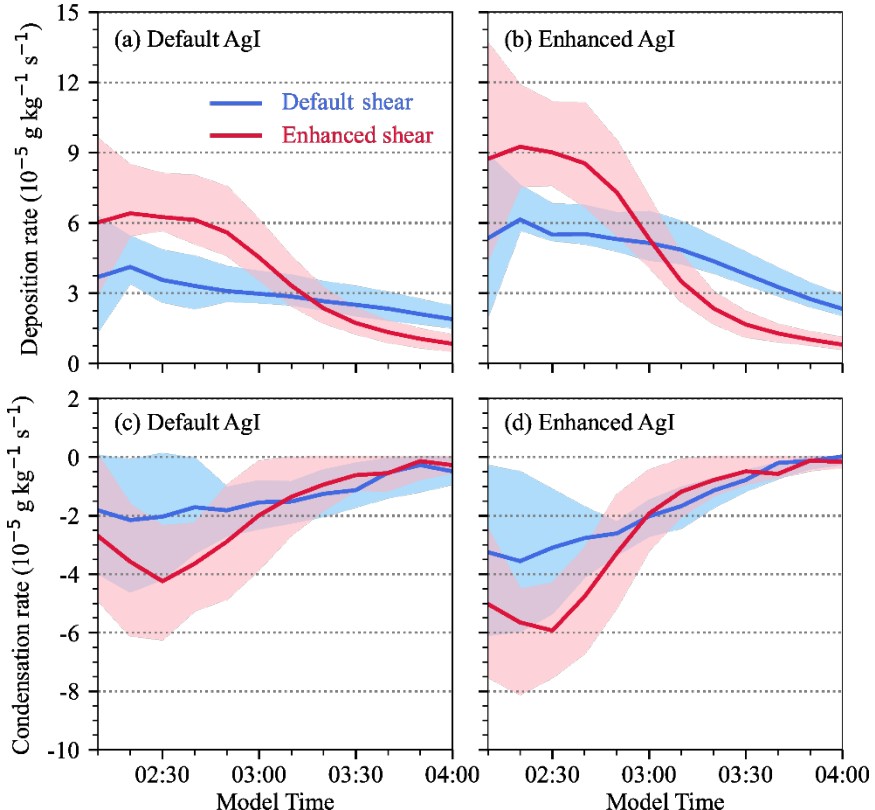

**Figure 9. Temporal variation of (a, b) ice depositional growth rate and (c, d) droplet condensation rate in SEED areas at different AgI concentrations and shear intensities. The upper and lower boundaries of the shaded areas indicate the 75th and 25th percentiles and the solid lines represent the mean values.**


## 3.3 Surface precipitation

Due to the faster cloud glaciation induced by stronger turbulence, it is expected that the precipitation may be enhanced only within a short time in the SEED areas after seeding is performed, and the seeding effect may be negative after the cloud is glaciated. Figure 10 shows the difference in the precipitation rates and the cumulative precipitation between the SEED and

NOSEED areas. Note that the surface precipitation rate was rather low because of the strong sublimation caused by the dry sub-cloud layer (Fig. 3). However, the differences in precipitation between the SEED and NOSEED areas are clear. It is seen from the figure that the positive seeding effect was significant after the turbulence was enhanced. The enhancement in precipitation rate in EnWS was even greater than that in the EnAgI experiment. With both turbulence and AgI enhanced, the maximum enchantment in precipitation rate was about 10 times greater than that in the Control experiment, resulting in greater

cumulative precipitation in the SEED areas (Fig. 10b). However, due to the fast depletion of liquid water, the precipitation rates in the EnWS and EnWS/EnAgI experiments were lower in the SEED area than in the NOSEED area after 4:10 MT. In





clouds with default turbulence intensity, the seeding effect was positive most of the time if using less AgI amount, and the cumulative precipitation generally increased with time. With more AgI seeded, the seeding effect became negative, and the cumulative precipitation decreased after 4:40 MT. Due to the slower cloud glaciation process, the seeding-induced enhanced cumulative precipitation in the EnAgI experiment exceeded that in the EnWS experiment (Fig. 10b).

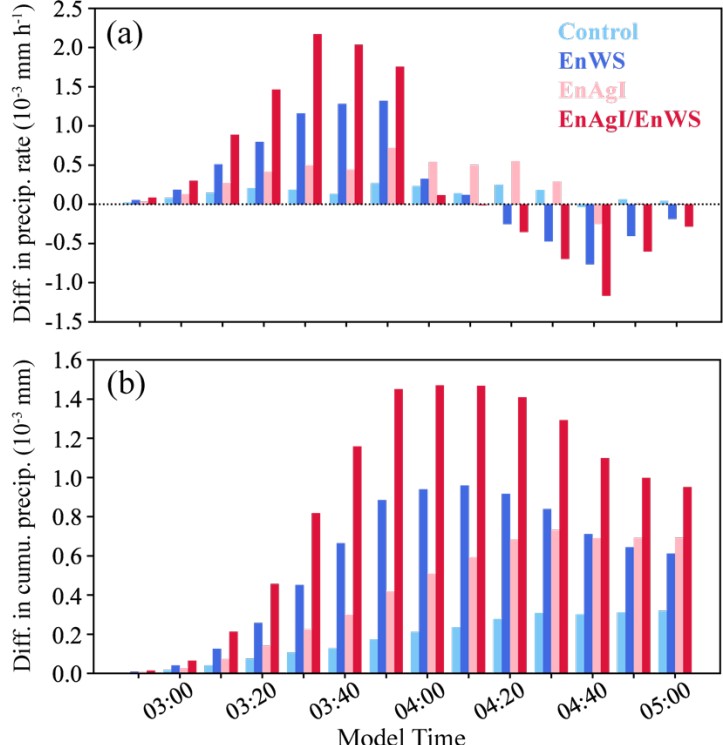

**Figure 10. Temporal variations of the differences in (a) precipitation rates and (b) cumulative precipitation between SEED and NOSEED areas.**

The negative seeding effect induced by turbulence after 4:10 MT resulted in a decrease in precipitation in the downwind SEED areas. This can be intuitively seen from the maps of differences in cumulative precipitation compared to the average natural precipitation (Fig. 11.). In the Control simulation (Fig. 11a), the SEED area obtained more precipitation than NOSEED areas all the time. With enhanced AgI concentration (Fig. 11c), the precipitation in the SEED area obtained more enhancement, especially between 40-50 km along the x-distance. However, as the cloud moved further downwind, the magnitude of the seeding effect became similar in the EnAgI and the Control experiments. With turbulence enhanced (EnWS), the accumulated precipitation increased rapidly in the SEED area and then decreased as the cloud moved northeastward. The seeding effect became negative in the downwind areas, indicating the presence of the "robbing Peter to pay Paul" effect. Such a transition from positive to negative seeding effects was more substantial in magnitude in the EnWS/EnAgI experiment than in the EnWS experiment (Fig. 11d), but the area with a negative seeding effect is similar to that with a positive effect in both simulations.



On the contrary, under weaker turbulence, even though the cumulative amount of precipitation enhanced by seeding is less than that with stronger turbulence, it generates more sustained precipitation enhancement and is more beneficial if one wants a positive seeding effect in a larger target area. The absolute (relative) increases in water volume in the areas affected by seeding (black boxes in Fig. 11) are 976.3m$^3$ (8.0%), 1291.2m$^3$ (5.1%), 2042.7m$^3$ (16.7%), and 2234.1m$^3$ (8.7%) in the Control, EnWS, EnAgI, and EnAgI/EnWS experiments, respectively.


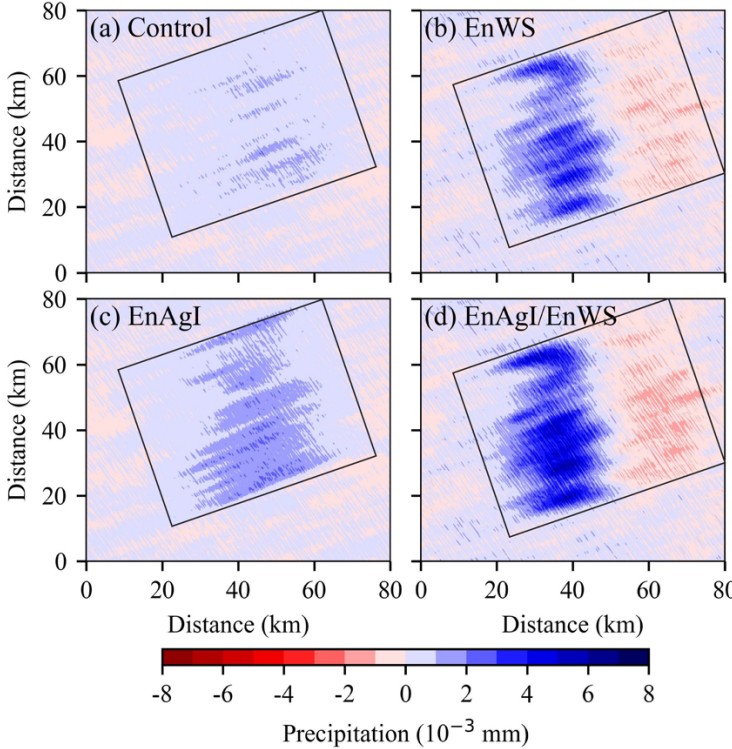

**Figure 11. Maps of difference in cumulative precipitation compared to the average natural precipitation from the (a) Control, (b) EnWS, (c) EnAgI, and (d) EnAgI/EnWS experiments.**

**3.4 Role of turbulence in cloud glaciation**

Turbulence helps the clouds maintain a mixed-phase state by enhancing mixing and liquid condensation, but it also promotes the dispersal and activation of AgI INP and the growth of ice crystals. For the case presented in this paper, the latter dominated in the SEED area. This section further quantifies the competition among turbulent mixing, liquid condensation, and cloud glaciation.






Figure 12 shows the characteristic times of turbulent mixing and cloud glaciation in the SEED area, which are calculated using the formulas in Korolev and Milbrandt (2022):

$$\tau_{mix} = \left(\frac{L^2}{\varepsilon}\right)^{1/3} \tag{1}$$

$$\tau_{gl} = \frac{\rho_i}{4\pi c S_i(T)}\left(\frac{9\pi}{2}\right)^{\frac{1}{3}}\left(\frac{1}{\rho_i}\right)^{\frac{2}{3}}\left(\frac{LWC}{N_i}\right)^{\frac{2}{3}}\left[\frac{L_i^2}{kR_v T^2} + \frac{R_v T}{E_i(T)D}\right] \tag{2}$$

where $L$ is the spatial scale in m, and $\varepsilon$ is the turbulence energy dissipation rate in m$^2$ s$^{-3}$, which is proportional to turbulent kinetic energy (e.g., Pokharel et al., 2017). $\rho_i$ is the density of ice in kg m$^{-3}$, $c$ is the ice particle shape factor characterizing capacitance ($0<c\leq1$, $c=1$ for sphere), $S_i(T) = \frac{E_w(T)}{E_i(T)} - 1$ is the supersaturation over ice, $E_w(T)$ and $E_i(T)$ are the saturation vapor pressure with respect to liquid and ice at temperature $T$, respectively. $N_i$ is the ice particle concentration, $L_i$ is the latent heat for ice sublimation, $k$ is the coefficient of air heat conductivity, $R_v$ is the specific gas constant of water vapor, and $D$ is
the coefficient of water vapor diffusion in the air.

In all our simulations, as the AgI particles disperse, the scales of the seeding plume broaden, resulting in a larger $L$. Therefore, the mixing time scale $\tau_{mix}$ increases with time, indicating more time is required to refill the SEED area with liquid water through mixing only. On the other hand, the cloud glaciation time scale $\tau_{gl}$ decreases with time because the LWC is
continuously reduced in the SEED areas (Figs. 8c, f, i, and l).

By comparing the right and left panels in Fig.12, it can be seen that the enhanced turbulence accelerated both cloud glaciation and turbulent mixing. Applying more AgI aerosols had negligible impacts on $\tau_{mix}$ but clearly enhanced the cloud glaciation (Fig. 12c). The intersection of the curves, which indicates the time when the rate of cloud glaciation exceeded the turbulent
mixing, was significantly advanced by the stronger turbulence and stronger seeding rate in EnAgI/EnWS (Fig. 12d). The differences among the four panels better explain why turbulence enhanced the cloud glaciation. In the Control experiment, in which the mixed-phase cloud maintains in the SEED area for a relatively long period, $\tau_{gl}$ is smaller than $\tau_{mix}$ after 3:00 MT, meaning that the turbulent mixing was not fast enough to fill the seeding plume with liquid water, and new liquid formation must be more important to maintain the cloud in mixed-phase after this time.





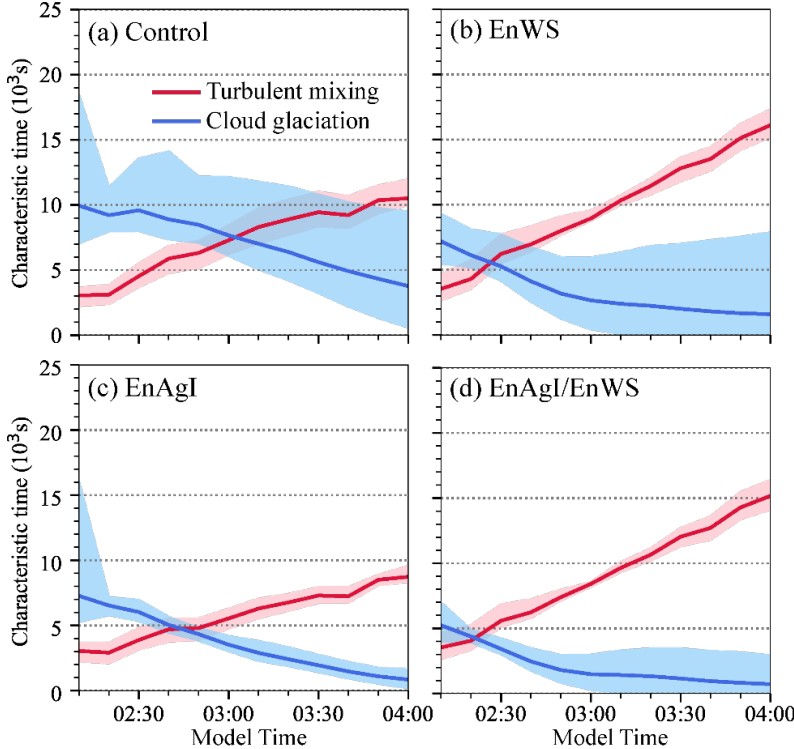

**Figure 12. Temporal variations of the characteristic times of turbulence mixing and cloud glaciation in SEED areas from the (a) Control, (b) EnWS, (c) EnAgI, and (d) EnAgI/EnWS experiments. The upper and lower boundaries of the shaded areas indicate the 75th and 25th percentiles and the solid lines indicate the means.**

Although the cloud glaciation was accelerated by turbulence, it does not indicate droplet condensation was entirely prevented in the SEED areas because there were updrafts strong enough to force droplets to grow. Korolev and Mazin (2003) proposed the threshold of vertical velocity ($w^*$) for which both droplets and ice can grow as

$$w^* = \frac{e_s - e_i}{e_i} \eta N_i r_i V_f \tag{3}$$

where $\eta$ is a coefficient dependent on temperature and pressure, $V_f$ is the ventilation factor, $N_i$ and $r_i$ are the number concentration and mean radius of ice crystals, respectively. In our simulations, condensation occurred in the area with a vertical velocity greater than $w^*$, while evaporation took place when the vertical velocity is smaller than $w^*$, regardless of whether turbulence or AgI amount were enhanced (Fig. 13). Therefore, liquid water condensation can still occur in SEED areas even after the cloud is glaciated. However, the fractional area with a positive condensation rate substantially decreased with time (Fig. 13c), especially in the experiments with enhanced turbulence. Liquid water that forms in such a small area can be rapidly consumed by ice growth and there is no chance for them to fill the glaciated area through turbulent mixing. Therefore, the cloud ultimately glaciated in SEED areas.

In short, for the case presented here, stronger turbulence can enhance the mixing and updrafts in which condensation can occur in SEED areas, but the mixing is too slow and the area where droplets can grow is too small to maintain the mixed-phase clouds, resulting in fast cloud glaciation and decrease in precipitation in downwind SEED areas.

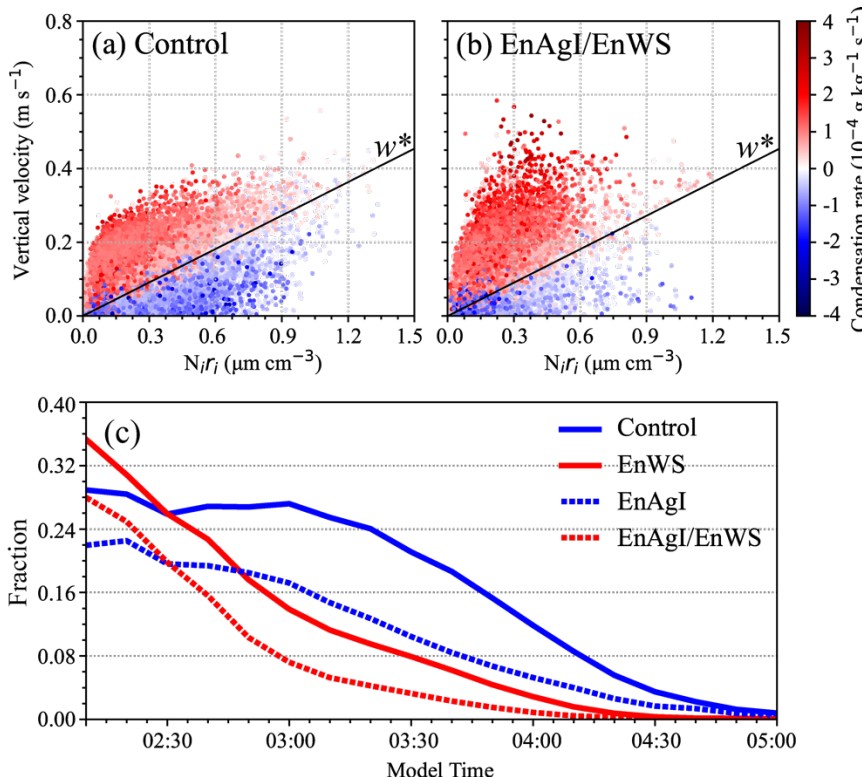

**Figure 13. Condensation rate in SEED areas under different vertical velocities and $N_i r_i$ from (a) Control and (b) EnWS/EnAgI experiments at 3:30 MT. The black lines are the minimum vertical velocity for which both liquid and ice can grow, which are calculated using Eq. 3 at 700 hPa and -15 °C. (c) The fraction of area with positive condensation rate in SEED areas.**

## 4 Discussions

In this study, the impacts of turbulence on airborne-based glaciogenic seeding effects are investigated using WRF-LES. The results show that for the case analyzed here, a well-capped, shallow (~500 m deep) decoupled stratus cloud with cloud top of -16°C, stronger shear-driven turbulence can enhance the dispersion of AgI particles and the nucleation and growth of ice crystals, which ultimately produces more precipitation but accelerates the cloud glaciation, even though the mixing and liquid condensation are also enhanced. These impacts are the same for seeding with a lower or a higher AgI aerosol amount. Such a negative downwind effect is also shown in some previous modelling studies. For example, Pourghasemi et al. (2022) coupled the aerosol-aware Thompson-Eidhammer microphysics scheme with an aerosol model and applied it to the WRF to simulate




airborne seeding in the upwind region. The seeding experiment was realized by varying the concentration of ice nuclei aerosols
after AgI had been dispersed in clouds. Compared to the experiment without seeding, the aerosol increase produced more
cloud water, more intense vertical airflow, and a 4.1% decrease in mean accumulated precipitation in the downwind area of
positive precipitation induced by seeding. In our study, an increase in AgI amount slightly reduced the downwind precipitation
(Fig. 10), but we highlighted that dynamics such as turbulence are also very important in controlling the downwind effect.

The results obtained in this study are based on a case study using idealized LES and a few sensitivity tests of different turbulent
intensities and AgI amounts, however, the basic principle should be the same for different cases: it is the competition among
liquid condensation, mixing and cloud glaciation determines whether turbulence helps to maintain the clouds in mixed-phase
or create the "robbing Peter to pay Paul" effect. We acknowledge that turbulence may play different roles in different cases.
For example, since ice growth is slower at temperatures colder or warmer than -15 °C (Yang et al., 2024b), turbulence may
have a weaker positive impact on ice growth, thus, it could be helpful to maintain the cloud in mixed-phase if seeding at a
temperature that is different from this study. In addition, we investigated a shallow stratiform cloud here. For deeper clouds
such as nimbostratus, which are often associated with frontal passages, there may be more liquid water supply and stronger
natural ice nucleation, including through secondary ice production processes. It is less likely that turbulence alone can result
in complete cloud glaciation in SEED areas in these clouds, at least there is no observational evidence. Moreover, the updrafts
and downdrafts in this study are only driven by turbulence, if turbulence is imposed on a larger-scale dynamic forcing (e.g.,
orographic gravity waves), it may have different impacts on cloud microphysics such as enhancing the riming and aggregation
(e.g., Houze and Medina, 2005; Grasmick and Geerts 2021; Grasmick et al. 2022) processes, which may in turn result in a
different impact on the seeding effect. Finally, if the layer with supercooled droplets is sufficiently close to moist neutrally
stratified, then the glaciation by airborne seeding may release sufficient heat to result in buoyant ascent, creating its own
turbulence, raising cloud top heights, and possibly enhancing surface precipitation, i.e., the dynamic seeding concept (Simpson
et al., 1967; Bruintjes et al., 1999). In the case simulated here, the cloud layer was too stably stratified for such a buoyant
ascent of seeded air parcels. To provide a complete understanding of how turbulence affects the glaciogenic seeding effect,
more observational and modelling studies are needed in the future.

We showed that the natural precipitation can be enhanced due to stronger turbulence, which is consistent with previous studies
that used different methods to alter the turbulence intensity in the model. For example, by using buoyancy perturbation to
induce turbulence in LES, Chu et al (2018) found that the net outcome of turbulence on snow growth is positive and leads to
a net increase in precipitation amount and duration. Recently, Sarnitsky et al. (2024), based on a statistical model, suggested
that submeter turbulence has minor impacts on ice growth, while turbulences on larger scales can strongly affect cloud
glaciation. In our simulation, we can only resolve the turbulence larger than 100 m, but the results of how turbulence affects
cloud phase partitioning and precipitation are similar to previous studies using finer resolution (Chu et al., 2018; Yang et al.,
2024a). Although we do not focus on the impacts of turbulence on natural precipitation in this study, the results provide



additional evidence that turbulence can significantly affect the precipitation in both NOSEED and SEED areas, thus, its impact on cloud microphysics and precipitation should be carefully parameterized in numerical weather prediction (NWP) models.

## 5 Conclusions

In this study, by using LES, we investigated the impacts of turbulence on airborne glaciogenic cloud seeding effect in mixed-phase stratiform clouds. The case was conducted over North China Plain, with complete cloud glaciation observed in the SEED area. The model well captured the magnitude and spatial distribution of seeding-induced reflectivity enhancement along the flight track. Four sensitivity tests were conducted, a control run which is driven by the default sounding data and actual seeding strategy, and the other three in which turbulence or (and) AgI release rate are enhanced. The turbulence is enhanced by intensifying the vertical wind shear. The main findings are as follows:

(1) Stronger turbulence can accelerate the seeding effect by enhancing AgI particle dispersion, ice nucleation, and ice growth through the WBF process, resulting in faster consumption of liquid water and cloud glaciation in the SEED area, even though stronger turbulence also enhances the liquid water formation.

(2) Once cloud glaciation is accelerated by stronger turbulence, the precipitation rate can be enhanced within a short time after seeding is performed, but the downwind precipitation may decline, causing a "robbing Peter to pay Paul" effect. Such a transition from a positive to a negative seeding effect is more substantial for a higher AgI release rate.

(3) It is the competition among liquid condensation, mixing, and cloud glaciation that determines the downwind effect of glaciogenic cloud seeding. For the case presented in this paper, neither the liquid condensation nor the turbulent mixing can overcome the cloud glaciation intensification by turbulence.

Although this study is based on case analysis with limitations, the results provide strong evidence that turbulence plays a vital role in the dynamical/ microphysical chain of events associated with glaciogenic cloud seeding. To deepen our understanding of the role of turbulence in natural and seeded clouds, more observational and modelling studies are needed in the future. In addition, to better simulate natural and seeded clouds and precipitation in NWP models, further development of parametrizations capturing the impact of turbulence on ice initiation and other mixed-phase cloud processes is needed.



**Data availability**

The WRF model is available at https://www2.mmm.ucar.edu/wrf/users/download/get_source.html (NCAR MMM, 2023). The sounding data, radar data, and satellite data are available at https://doi.org/10.5281/zenodo.14604420 (Yang, 2025).

**Author contributions**

MC, XJ, and JL conducted the numerical simulations. MC, XJ, and JY analyzed the observational and model results. MC, XJ, and JY prepared the paper. XD provided the data of radar measurements. BG, YY, BC, and XL provided inputs on the method and analysis. All the authors provided significant feedback on the paper.

**Competing interests**

The contact author has declared that none of the authors has any competing interests.

**Acknowledgments**

This work was supported by the National Natural Science Foundation of China (42475201, 42230604), and the CMA Key Innovation Team Support Project (CMA2022ZD10). The authors acknowledge the High Performance Computing Center of Nanjing University of Information Science & Technology for their support of this work, and we acknowledge the Xingtai Atmospheric Environment Field Scientific Test Base of CMA for collecting the data and for providing high-quality products.
We appreciate the editor and reviewers for their insightful comments and suggestions.

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
