# Peer review of "Accelerated impact of airborne glaciogenic seeding of stratiform clouds by turbulence"

_EGUsphere, 2025_

## Referee Comment (RC1)

**Review for "Accelerated impact of airborne glaciogenic seeding of stratiform clouds by turbulence" by Meilian Chen, Xiaoqin Jing, Jiaojiao Li, Jing Yang, Xiaobo Dong, Bart Geerts, Yan Yin, Baojun Chen, Lulin Xue, Mengyu Huang, Ping Tang, and Shaofeng Hua**

The manuscripts discusses a cloud seeding case study used for investigating the effect of turbulence on the dispersion of the seeding plume, the ice nucleation, ice crystal growth, and subsequent precipitation formation. It employs idealized WRF LESs to quantify the effect of turbulence. The authors provide a nice overview of the case study and their methods. Their findings are interesting for the scientific community focusing on mixed-phase clouds. In general, the manuscript is well written using clear language and the figures are well done. I do have comments regarding study setup and scientific conclusions. After addressing them, I recommend the manuscript for publication.

**Major comments**

- Throughout the manuscript, different time scales are discussed (e.g., glaciation time and turbulent mixing). In lines 34-37, the complex interplay of cloud seeding effects, dynamics, and microphysics is mentioned. It would be helpful if the authors can spend more time on that, as this is at the heart of their study. How are dynamics and turbulence separated? The idealized simulations employ a horizontal resolution of $100\,\mathrm{m}$. Is this enough to resolve the largest eddies? How does the definition of turbulence influence the interpretation of the results? The authors often talk about complete cloud glaciation, but what does that mean? The seeding plume has no LWC anymore? What is the definition of that?

- The case study description should be extended by more numbers already in the text. The authors say the seeding area was characterized by high RHs, from Figure 1a it looks like 50-60 %, which in my opinion is not high. It is further stated that the cloud is decoupled form the surface, $500\,\mathrm{m}$, and with a cloud top temperature of -16 °C. Where is this information coming from? From Figure 2 I can only see that the radar reflectivity is low, i.e., that most likely it was non precipitating. The authors should avoid using the jet colormap (Figure 1, continuous or discretized) given the known issues with using that colormap (this goes for all figures having that colormap). I further cannot see the -16 °C cloud top temperature given the scale only goes to -14.5 in Figure 1d. What is the reason for choosing such a large range given that most of the observed temperatures were in-between -14 and -12 °C. In Figure 1, is the pink area in a and b corresponding to the view in c and d? If not, it could help to do it that way.

- How do the authors define NOSEED and SEED areas? Why wasn't a simulation with no seeding used as a reference for no seeding effect? This is important to know, especially when the authors talk about the change in precipitation.

- A short description on the natural ice nucleation in the model should be done. What parameterization is active at these temperatures and how strong is the freezing to be expected? This should also be discussed in relation to the observed case, where basically no ice nucleation (low radar reflectivity) was observed. So, is it relevant to turn on ice nucleation in the model? This is also of importance to the discussion about the effect of turbulence on ice nucleation.

- Radar reflectivity and cloud thickness in the model simulations: Figure 5 shows a rather thin liquid cloud layer with strong seeding signals. I wonder what changed in the radar reflectivity in Figure 7 as there is a stronger signal in the seeding, but also from the background cloud? Also is the cloud thickening from 2:30 to 3:00 in Figure 7? I am further confused by the signal of LWC (cloud water mixing ratio) in Figure 8 as it appears to be more widespread than in Figure 5 (over more vertical levels). Am I confusing here something? A difference plot with a reference simulation (no seeding) could also be helpful to disentangle turbulence induced by the seeding plume and from the environment.

- Precipitation: The results are presented in a nice way, but I am missing a proper discussion on the significance of the simulations (i.e., uncertainty) and especially the scale is omitted in the text. The figures show a scale of 0.1 $\mu\mathrm{m\,h^{-1}}$ of changes in precipitation? Also here the computation of the differences in SEED and NOSEED should be made clear (see comment above). Figure 11: the boxes seem to be of different sizes and I am wondering what the numbers in the text really tell me given that the colors show the precipitation and not total water volume.

**Minor comments**

- Line 17: the acronym LESs does not have to be introduced as it is not used in the abstract

- Line 18: "the model can reasonably capture" - is there a word missing? reasonably well?

- Line 29: "similar crystal structure" to "similar molecular lattice structure"

- Line 30: Here the onset freezing temperature for AgI should be stated and also other references, such as Marcolli et al., 2016 and Chen et al., 2024 should be included.

- Line 41: What is the result of complete cloud glaciation? The decrease in cloud top and complete cloud clearing? Again here the definition of complete cloud glaciation is needed. I further do not understand the second part of the sentence, is the mixing consuming water or is it producing more through cloud droplet formation / growth?

- Line 59: definition of plains should be stated earlier, as it is already used before in the introduction and in the abstract

- Line 63: The formula by Korolev and Mazin is later used, maybe this can be pointed out here, otherwise this reads as a rather random information. This also can be said for the next sentence, regarding the findings by Korolev and Field (2008). What are the authors trying to convey with this information? How is this relevant for their study?

- Line 68: the introduction of LWC is good and should be used throughout the manuscript (instead of liquid water and cloud water mixing ratio). This way, it is consistent and will help ease the understanding.

- Line 70: Does "suppress this through cloud top entrainment" refer to ice growth or / and precipitation?

- Line 79: I think it is valid to conduct the study over flat lands, but the authors say it is also relevant to mountainous terrain. Here, more justification for this interpretation would be great.

- Line 91: already here the cloud type (i.e., stratiform) can be defined.

- Line 156: Where is the sounding coming from? Is this a real observation? Or did the authors prescribe an artificial profile (also fine)?

- Figure 3: Can you add the RH profile, such that the conditions are more easy to grasp? Especially when you talk about dry intrusions from above, this could be helpful instead of having to do the calculations oneself.

- Line 203: What is the resolution of the radar?

- Line 206: A reference to Omanovic et al., 2024 should be done, given that they reported a similar result with too slow WBF process.

- Line 241: "scales of seeding plumes"? Do you mean the vertical extent? The spread?

- Line 260: What do you mean by similar variations? In terms of magnitude? Pattern?

- Line 265: I believe difference plots between the simulations would help here the discussion, as the difference can be quite subtle and I cannot follow the discussion of the authors on the changes.

- Line 275: I understand if you do not want to dive into riming and aggregation, but could you provide more information on that? Maybe an appendix figure? Riming and aggregation should occur especially with these high ice concentrations.

- Line 279: I believe there is a word missing before "became lower ..." or what do the authors refer to? The deposition rate?

- Line 309: enchantment to enhancement?

- Line 362: I might understand Figure 12 wrong, but enhanced turbulence does not enhance both cloud glaciation and turbulent mixing. I mostly see a reduction in the time for cloud glaciation, while for turbulent mixing this is more difficult to see. Maybe a quantification could be helpful here.

- Figure 13: How can you have negative condensation rates (evaporation) with vertical velocities larger than $w^*$? Can you quantify how often you encounter which conditions, i.e., both growing or only ice crystals growing?

- Line 397-401: this is one sentence, can you split it up?

- Line 414: There are other (older) reference to ice crystal growth across temperatures, please add them.

- Line 417: Nimbostratus produce natural precipitation. Why did you choose this cloud type as an example? Is it really ideal to be seeded?

- Line 448: What ways do you see to parameterize turbulence in connection to cloud microphysics? Are there open questions in this regards?

**References**

Chen, J., Rösch, C., Rösch, M., Shilin, A., & Kanji, Z. A. (2024). Critical Size of Silver Iodide Containing Glaciogenic Cloud Seeding Particles. *Geophysical Research Letters*, *51*(7), e2023GL106680. https://doi.org/10.1029/2023GL106680

Marcolli, C., Nagare, B., Welti, A., & Lohmann, U. (2016). Ice nucleation efficiency of AgI: Review and new insights. *Atmospheric Chemistry and Physics*, *16*(14), 8915–8937. https://doi.org/10.5194/acp-16-8915-2016

Omanovic, N., Ferrachat, S., Fuchs, C., Henneberger, J., Miller, A. J., Ohneiser, K., Ramelli, F., Seifert, P., Spirig, R., Zhang, H., & Lohmann, U. (2024). Evaluating the Wegener–Bergeron–Findeisen process in ICON in large-eddy mode with in situ observations from the CLOUDLAB project. *Atmospheric Chemistry and Physics*, *24*(11), 6825–6844. https://doi.org/10.5194/acp-24-6825-2024

---

## Referee Comment (RC2)

**Title:** Accelerated impact of airborne glaciogenic seeding of stratiform clouds by turbulence

**Manuscript Number:** EGUSphere-2025-47

**Recommendation:** Accept after major revisions

Chen et al. investigates how turbulence modulates the effects of AgI cloud seeding in shallow stratiform clouds over flat terrain. Using LES simulations based on a real seeding event over North China on 20 January 2022, this study creates an artificially enhanced shear layer and finds that increased turbulence enhances AgI dispersion, ice nucleation, and crystal growth, which accelerates faster glaciation. Although stronger turbulence fosters enhanced SLW generation, the rate of ice growth outpaces condensation leading to a rapid depletion of SLW and faster transition to glaciated cloud. Turbulence amplifies short term precipitation directly downwind of seeding, it suppresses further downwind precipitation. This study highlights the delicate balance between turbulent mixing and glaciation in determining the spatial extend and efficiency of cloud seeding impacts. Overall, I felt like the study was well written but could benefit from several clarifications to enhance the study. Therefore, I am recommending major revisions.

**General Comments:**
In Fig. 1, the visible and infrared imagery show a clear depression in the cloud top along the flight track. However, it would be beneficial to demonstrate whether this displacement is consistent with the large-scale flow pattern. Specifically, could the flight track be transposed or advected downwind using the wind vectors from the ERA5 reanalysis fields shown in Fig. 1a–b? This would better contextualize the observed cloud top depression and support the attribution of the cloud signature to seeding effects.

Similarly, in Fig. 2, the location of the S-band radar relative to the flight track is unclear and should be explicitly marked on a map. The reflectivity evolution shows a clear enhancement along one of the flight legs but not the other—why does the enhanced reflectivity from the eastern leg fade later in time? Was there a difference in cloud properties, wind shear, or moisture that could have contributed to the disparity between the seeded legs?

The choice of background aerosol concentration ($N_0 = 4000$ $cm^{-3}$) seems quite high for a wintertime stratiform cloud and may not be representative of the conditions aloft over rural North China. Is there observational support for this aerosol profile? Were vertical variations in aerosol concentration considered, and how sensitive are the results to these assumptions? Given that aerosol background can strongly influence microphysical pathways, this deserves more justification or discussion.

A brief explanation of how cloud was forced and maintained in the model would benefit the reader. While the use of a constant profile and a single sounding is mentioned (Section 2.2), it was not immediately clear how the cloud was maintained within the LES framework, particularly given the absence of evolving large-scale forcing. I found myself needing to read through this section twice to piece together how the cloud structure was sustained. A concise summary early in the model setup section—perhaps explicitly stating the role of the initial liquid water mixing ratio, the

static thermodynamic profile, and discussing how cloud was maintained after it was initialized—would improve clarity.

The LES is driven by a single sounding with no horizontal heterogeneity or time-evolving large-scale forcing. This idealized setup limits realism, especially in terms of representing synoptic evolution and mesoscale variability. How might the results differ if the full 3D ERA5 meteorological fields were used to initialize and force the model? Would the cloud form in a similar vertical structure, and would radar reflectivity fields appear more similar to observations? A brief discussion of this limitation would improve transparency.

One of the more confusing aspects of the paper is the comparison setup (especially compared to other seeding studies). The so-called "Control" simulation includes AgI seeding, but there is no true baseline simulation without any seeding. This makes it difficult to isolate the seeding effect, especially since fall streaks and reflectivity enhancements appear in the Control run, which undermines the attribution of microphysical or dynamical responses to seeding. Including a no-seeding case—or clearly differentiating it from the "Control"—would clarify the results and interpretation. I think running a true control and comparing it to the observations and other simulations would add a lot of value to the manuscript.

The simulations use only a one-hour spin-up period before seeding is introduced at 2:00 MT. However, at 2:30 MT (the first output shown in Fig. 7), the cloud field and reflectivity structure still appear immature. For instance, in all simulations including the Control, reflectivity increases dramatically in later time steps. This raises the possibility that early reflectivity enhancements are a spin-up artifact rather than a response to seeding. A longer spin-up or justification for the current choice is needed, particularly since the cloud system being simulated is shallow and sensitive to small thermodynamic changes. Also, in Fig. 7, the radar reflectivity differences between SEED and NOSEED areas are subtle, especially in the early time steps. The seeding signature is difficult to identify without annotations or overlays. Annotating the plots to highlight the SEED plume location, expected signal from seeding, and NOSEED comparison regions would make the figures much more interpretable. As it stands, the reader must infer these spatial relationships without guidance.

While the discussion offers some thoughts on generalizability, the study's findings are drawn from a highly specific case: a shallow, capped stratiform cloud in a quiescent wintertime environment. The conclusions regarding turbulence-enhanced glaciation and the transition from positive to negative seeding effects may not extend to deeper, more dynamic clouds. A more tempered framing of the conclusions, emphasizing the case-specific nature of the results, would strengthen the overall presentation

**Specific Comments:**
**Line 103** – "...brightness temperature increased by about 2 °C..." Is this value within the noise level of the satellite product? What's the uncertainty?

**Line 116** – "...radar echo appeared about ten minutes after seeding..." Is this consistent with modeled ice nucleation onset time? If not, why not? Feel like this was not discussed later in the text

**Line 135**: "...periodic lateral boundaries..." Could periodic boundaries introduce artifacts in this stratiform cloud case?

**Line 158** – "A single sounding is used to drive the model…" Can the authors clarify when and where the sounding was launched relative to the seeding flight track and time? Was it an operational sounding, and how representative is it of the cloud environment during the seeding event? Spatial and temporal offsets could influence the initial conditions and evolution of the simulated cloud.

**Line 164**: "...Richardson number...from 16.81 to 0.67..." This is a large reduction. Can you show whether turbulence actually became dynamically unstable (e.g., Ri < 0.25)?

**Line 265**: "...IWC up to 0.06 g/kg..." How does this compare with in-situ observations from similar seeded clouds?

**Line 286**: "...generation of liquid water was significantly slower..." Can this be quantified more directly using a rate ratio (e.g., production/consumption)?

**Line 311**: "...cumulative precipitation decreased after 4:40 MT..." What was the baseline precipitation without seeding during this period?

---

## Referee Comment (RC3)

Review of

Accelerated impact of airborne glaciogenic seeding of stratiform clouds by turbulence

Meilian Chen et al.

DOI 10.5194/egusphere-2025-47

24 March 2025

Glaciogenic cloud seeding is studied through numerical simulations of a case study of a shallow, stratiform cloud comprised of supercooled liquid water. WRF LES is employed for these simulations with a fast spectral bin microphysics scheme. Four simulations are undertaken to test the relative importance of turbulence generated by vertical winds shear (control vs enhanced) and the concentration of the released AgI seeding agent (control vs enhanced).

The primary conclusion highlights the importance of enhanced turbulence in accelerating glaciogenic seeding in comparison to enhanced AgI concentrations.

The paper is, in general, well-written with appropriate figures laid out in the logical manner. The literature review/introduction was particularly well done. I did find the writing to be fairly repetitive, especially with respect to the effect of turbulence through the results, enough so that I am specifically commenting about it. At other points, I thought some material was missing, as detailed below.

I respect and appreciate that the author readily acknowledges the inherit sensitivity of weather modification to the specific meteorology/event, what works in this case study may not work for a different event.

The majority of my comments pertain to further clarification of the work and assumptions rather than making new simulations. As such I consider the comments to be minor in nature.

Comments:

Why do you want to run in two different times, with the case study being in UTC and the simulations being in model time? This made it more difficult to go back and forth between the two. Does seeding at 2:00 MT correspond to ~ 8:00 UTC? Does this mean the simulations are spun up for two hours, rather than one? Given that 8:45 UTC is 16:45 CST, would the local time be around local dusk for January, which is why the visible image has no colour?

Figure 1: Given that the simulations only go up to 3 km, I'd rather see the MSLP and surface temperature rather than the 500 hPa geopotential height. In particular, I'd like to know more about the boundary layer stability and boundary layer dynamics, beyond "quiescent and stably stratified". Do we have warm air advection or cold air advection? The satellite

imagery suggests that this solid SLW cloud layer is moving up from the south, as discussed in the manuscript.

Figure 3: When and where was the sounding taken?  Was this from the seeding aircraft or an upper air sounding?

I am confused between Figure 5, which has a steady cloud comprised of SLW, and Figure 7, which has ice going to the surface and Figure 8, which has the cloud water mixing ratio diminishing through the simulation.  That is directly at odds with Figure 5.  I assume that this is a consequence of this being 'the control simulation without natural ice simulation' while Figures 7 & 8 allow natural ice nucleation.  This needs to be explained in much greater detail.  Would a true 'control' run (i.e., no enhanced shear and no seeding at all) glaciate just like the NOSEED areas in Figure 7?  Is the reflectivity in Figure 7 being caused by ice failing below the cloud deck?  I assume this does not match reality.  What does this suggest about the formation of ice in the simulations?  It would be best not to gloss over this.

Line 245: magenta lines?  The magenta line is at the inversion and is clearly visible at 4:00 MT.

I am 90% confident that SEED area refers to the two primary red streaks in Figure 7 and NOSEED refers to everything else.  I think, however, a couple of arrows pointing to what you are calling SEED would be very helpful.

Given that there is no source of water (latent heat flux) in the simulations, it seems obvious to me that any enhancement of precipitation upwind will require less precipitation at some point downwind.  (I.e., robbing Peter to pay Paul.) This is just a statement of the conservation of water.   If the simulation was run longer and the domain was bigger, I would expect to see the same thing for the non-shear cases.  Am I mistaken?  Ultimately, I don't think you can really address this questions using LES simulations with periodic boundary conditions.  Once the precipitation reaches the surface, it's gone.

---

## Author Comment (AC1)

The reviewer's comments are in black, and responses are in blue.

The manuscript discusses a cloud seeding case study used for investigating the effect of turbulence on the dispersion of the seeding plume, the ice nucleation, ice crystal growth, and subsequent precipitation formation. It employs idealized WRF LESs to quantify the effect of turbulence. The authors provide a nice overview of the case study and their methods. Their findings are interesting for the scientific community focusing on mixed-phase clouds. In general, the manuscript is well written using clear language and the figures are well done. I do have comments regarding study setup and scientific conclusions. After addressing them, I recommend the manuscript for publication.

Reply: We appreciate your insightful comments. The paper has been revised accordingly and has been improved. Please see our point-by-point response below.

Major comments:

Throughout the manuscript, different time scales are discussed (e.g., glaciation time and turbulent mixing). In lines 34-37, the complex interplay of cloud seeding effects, dynamics, and microphysics is mentioned. It would be helpful if the authors can spend more time on that, as this is at the heart of their study. How are dynamics and turbulence separated? The idealized simulations employ a horizontal resolution of 100m. Is this enough to resolve the largest eddies? How does the definition of turbulence influence the interpretation of the results? The authors often talk about complete cloud glaciation, but what does that mean? The seeding plume has no LWC anymore? What is the definition of that?

Reply: We appreciate your comment. In lines 34-37, the statement points out that if one wants to see an unambiguous seeding signature, the seeding-induced cloud phase change should be faster than the changes in natural variability (caused by either dynamics or microphysics). This is added to the paper. Actually, the whole paper is about this, focusing on cloud glaciation after seeding and the impacts of turbulence. For the case presented here, both the enhanced AgI amount and turbulence accelerate the phase change (liquid to ice), and it is faster than the changes in natural variability, so the seeding signature is unambiguous. Dynamics includes both turbulence and that has larger scales; in this study, we focus on turbulence, and the updrafts and downdrafts are only forced by turbulence. The idealized simulation with a resolution of 100 m can

resolve eddies larger than 600 m, it is sufficient to reveal the influence on the seeding effect based on our analysis in the paper. The impacts of smaller eddies on ice growth and seeding effects need to be further investigated in the future. This information is added to the paper. Complete glaciation means the liquid water in cloud regions that are affected by seeding is completely consumed by ice growth, resulting in the LWC approaching zero. This happens when the liquid water consumption by ice growth is faster than liquid water formation by dynamic forcing or liquid water supply from areas outside of the seeding region by turbulent mixing.

The case study description should be extended by more numbers already in the text. The authors say the seeding area was characterized by high RHs, from Figure 1a it looks like 50-60%, which in my opinion is not high. It is further stated that the cloud is decoupled form the surface, 500m, and with a cloud top temperature of-16°C. Where is this information coming from? From Figure 2 I can only see that the radar reflectivity is low, i.e., that most likely it was non precipitating. The authors should avoid using the jet colormap (Figure 1, continuous or discretized) given the known issues with using that colormap (this goes for all figures having that colormap). I further cannot see the -16°C cloud top temperature given the scale only goes to -14.5 in Figure 1d. What is the reason for choosing such a large range given that most of the observed temperatures were in-between-14 and -12°C. In Figure 1, is the pink area in a and b corresponding to the view in c and d? If not, it could help to do it that way.

Reply: We appreciate your comment. The RH in Fig. 1 is from the reanalysis data, it may not reveal the actual RH in clouds, but it is evident that the RH at 850hPa is relatively higher in the flight areas, while the RH is low at higher levels. The information about the cloud is from the sounding measurement (Fig. 3), this has been added to the revised paper. The jet colormap is changed to viridis. The cloud top temperature is from the sounding measurement (Fig. 3a and b), this is added in the text. The color range in Fig. 1d is revised accordingly. Yes, the pink areas in Fig. 1a and b correspond to the views presented in Fig. 2c and d, which include the actual seeding trajectory.

How do the authors define NOSEED and SEED areas? Why wasn't a simulation with no seeding used as a reference for no seeding effect? This is important to know, especially when the authors talk about the change in precipitation.

Reply: We appreciate your comment. SEED and NOSEED are defined as the areas affected and unaffected by the seeding plumes at each moment, respectively. This information is added to the paper. In this study, the field measurement had seeding performed, and we primarily focus on the impact of turbulence on the cloud seeding effect, so we use the simulation with actual AgI amount as the Control. Since the seeding signature is unambiguous, analysis between SEED and NOSEED areas inherently provides the necessary contrast between seeding and no seeding simulations.

A short description on the natural ice nucleation in the model should be done. What parameterization is active at these temperatures and how strong is the freezing to be expected? This should also be discussed in relation to the observed case, where basically no ice nucleation (low radar reflectivity) was observed. So, is it relevant to turn on ice nucleation in the model? This is also of importance to the discussion about the effect of turbulence on ice nucleation.

Reply: We appreciate your comment. We use the default natural ice nucleation parameterizations in the fast SBM scheme, including deposition and condensation nucleation, contact nucleation (Meyers et al., 1992), and immersion freezing (Bigg, 1953). The ice concentration generated from natural ice nucleation can be found in Fig. 8a (black contours), which is an order of magnitude lower than the ice concentration induced by seeding. Given that the cloud in the case is liquid, the natural ice nucleation was intentionally turned off in the simulation to compare with observation for model evaluation. However, turning on the natural ice nucleation in the subsequent experiments was primarily to better understand how turbulence affects natural ice generation and growth. After the turbulence is enhanced, the natural ice concentration is enhanced by about 2 times (black contours in Fig. 8d and j).

Radar reflectivity and cloud thickness in the model simulations: Figure 5 shows a rather thin liquid cloud layer with strong seeding signals. I wonder what changed in the radar reflectivity in Figure 7 as there is a stronger signal in the seeding, but also from the background cloud? Also is the cloud thickening from 2:30 to 3:00 in Figure 7? I am

further confused by the signal of LWC (cloud water mixing ratio) in Figure 8 as it appears to be more widespread than in Figure 5 (over more vertical levels). Am I confusing here something? A difference plot with a reference simulation (no seeding) could also be helpful to disentangle turbulence induced by the seeding plume and from the environment.

Reply: We appreciate your comment. The Control experiment without natural ice nucleation is shown in Fig. 5, which conforms to the actual conditions of the case. Fig. 7 includes all experiments listed in Table 1, where the background radar reflectivity is the result of natural ice nucleation. The cloud is not thickening, and the changes in radar reflectivity indicate the falling snow (below the cloud base). In Fig. 8, the vertical axis for LWC is presented on the right side (1.3-1.9 km height). We are sorry for the misunderstanding. It is now clarified in the figure caption.

Precipitation: The results are presented in a nice way, but I am missing a proper discussion on the significance of the simulations (i.e., uncertainty) and especially the scale is omitted in the text. The figures show a scale of 0.1 $\mu m\ h^{-1}$ of changes in precipitation? Also here the computation of the differences in SEED and NOSEED should be made clear (see comment above). Figure 11: the boxes seem to be of different sizes and I am wondering what the numbers in the text really tell me given that the colors show the precipitation and not total water volume.

Reply: We appreciate your comment. Yes, the precipitation rate variation presented in Fig. 10a is in $\mu m\ h^{-1}$ because the precipitation was quite weak in this study due to the strong sublimation above the surface (Fig. 7). This is true in observation as almost no surface precipitation was detected after seeding. The calculation methodology for Fig. 10 is described as follows: 1) Identify precipitation regions for SEED and NOSEED at each moment based on cumulative precipitation characteristics; 2) Computing the average precipitation for both regions; 3) Calculating the differences in precipitation rates and cumulative precipitation between SEED and NOSEED areas at each moment.

The boxes indicate the approximate SEED areas; they are different between the left and right panels. The numbers in the text present the changes in cumulative precipitation volume within the SEED areas of the four experiments, which are calculated by comparing the average cumulative precipitation inside and outside of the boxes. It is

seen from Fig. 11 that the precipitation changes induced by seeding and enhanced turbulence are quite clear; it is unlikely that model uncertainties can explain such a significant precipitation change. However, we agree that the model uncertainty can affect the magnitude of precipitation changes, such as that from the microphysics scheme and the unresolved smaller turbulent eddies. This discussion is added in Section 4.

Minor comments:

Line 17: the acronym LESs does not have to be introduced as it is not used in the abstract.

Reply: We appreciate your comment. The acronym LESs has been removed from the abstract.

Line 18: "the model can reasonably capture"- is there a word missing? reasonably well?

Reply: We appreciate your comment. "the model can reasonably capture" is changed to "the model can reasonably well capture".

Line 29: "similar crystal structure" to "similar molecular lattice structure"

Reply: We appreciate your comment. "similar crystal structure" has been changed to "similar molecular lattice structure".

Line 30: Here the onset freezing temperature for AgI should be stated and also other references, such as Marcolli et al., 2016 and Chen et al., 2024 should be included.

Reply: We appreciate your comment. The onset freezing temperature of AgI particles (-4 °C for AgI particles in 0.1 µm, and -8 °C for AgI particles in 1 µm) is added in the text. The references are added.

Line 41: What is the result of complete cloud glaciation? The decrease in cloud top and complete cloud clearing? Again here the definition of complete cloud glaciation is needed. I further do not understand the second part of the sentence, is the mixing consuming water or is it producing more through cloud droplet formation / growth?

Reply: We appreciate your comment. Complete glaciation means the liquid water in cloud regions that are affected by seeding is completely consumed by ice growth. "Mixing" here means "turbulent mixing", which indicates supercooled water in adjacent areas can mix into the glaciated cloud areas, leading to a mixed-phase state. Sorry for any misunderstanding. The sentence has been revised to: *"It is a result of complete glaciation in seeding areas, which means the liquid water consumption by ice growth is faster than liquid water formation by dynamic forcing or liquid water supply from areas outside of the seeding region by turbulent mixing. This leads to the liquid water content (LWC) approaching zero"*

Line 59: definition of plains should be stated earlier, as it is already used before in the introduction and in the abstract.

Reply: We appreciate your comment. The definition of plains has been added at line 45: *"decrease in cloud top has been reported in several studies in which seeding experiments were conducted over flat land (plains)"*

Line 63: The formula by Korolev and Mazin is later used, maybe this can be pointed out here, otherwise this reads as a rather random information. This also can be said for the next sentence, regarding the findings by Korolev and Field (2008). What are the authors trying to convey with this information? How is this relevant for their study?

Reply: We appreciate your comment. The sentence has been revised to: *"Korolev and Mazin (2003) proposed a formula (shown later in Section 3.4)…"*. Korolev and Field (2008) showed that turbulence is the key process to maintain the cloud in a long-lived mixed-phase state. This is relevant to this study because we aim to investigate whether turbulence can keep the cloud in a mixed-phase state after seeding or not.

Line 68: the introduction of LWC is good and should be used throughout the manuscript (instead of liquid water and cloud water mixing ratio). This way, it is consistent and will help ease the understanding.

Reply: We appreciate your comment. "LWC" has been appropriately applied in the revised paper.

Line 70: Does "suppress this through cloud top entrainment" refer to ice growth or / and precipitation?

Reply: We appreciate your comment. "suppress this through cloud top entrainment" refers to "suppress ice growth through cloud top entrainment". This has been revised in the paper.

Line 79: I think it is valid to conduct the study over flat lands, but the authors say it is also relevant to mountainous terrain. Here, more justification for this interpretation would be great.

Reply: We appreciate your comment. Relevant justification for this interpretation has been added to the paper. Turbulence also plays a vital role in particle dispersion and ice growth in orographic clouds (Xue et al., 2014; Chu et al., 2018; Jing et al., 2016)

Line 91: already here the cloud type (i.e., stratiform) can be defined.

Reply: We appreciate your comment. The cloud type (stratiform) has been added to this sentence.

Line 156: Where is the sounding coming from? Is this a real observation? Or did the authors prescribe an artificial profile (also fine)?

Reply: We appreciate your comment. The sounding comes from real observation. This information has been added to the paper.

Figure 3: Can you add the RH profile, such that the conditions are more easy to grasp? Especially when you talk about dry intrusions from above, this could be helpful instead of having to do the calculations oneself.

Reply: We appreciate your comment. The RH profile has been added to the Fig. 3.

[Figure]

Fig. R1. The initial vertical profiles of (a) temperature and potential temperature, (b) actual vapor mixing ratio, saturation vapor mixing ratio relative to water and relative humidity, and (c) original and enhanced U and V components. The grey shaded area in (b) indicates the initial liquid water mixing ratio.

Line 203: What is the resolution of the radar?

Reply: The range resolution is 250 m; this information is added to the paper.

Line 206: A reference to Omanovic et al., 2024 should be done, given that they reported a similar result with too slow WBF process.

Reply: We appreciate your comment. The reference has been added to the paper. Omanovic, N., Ferrachat, S., Fuchs, C., Henneberger, J., Miller, A. J., Ohneiser, K., Ramelli, F., Seifert, P., Spirig, R., Zhang, H., & Lohmann, U. Evaluating the Wegener–Bergeron–Findeisen process in ICON in large eddy mode with in situ observations from the CLOUDLAB project. Atmospheric Chemistry and Physics, 24(11), 6825–6844. https://doi.org/10.5194/acp-24-6825-2024, 2024.

Line 241: "scales of seeding plumes"? Do you mean the vertical extent? The spread?

Reply: Yes, it means horizontal spread. This is revised in the paper.

Line 260: What do you mean by similar variations? In terms of magnitude? Pattern?

Reply: The similar variations mean the variation pattern. The relevant sentence has been modified.

Line 265: I believe difference plots between the simulations would help here the discussion, as the difference can be quite subtle and I cannot follow the discussion of the authors on the changes.

Reply: We appreciate your comment. Since we also want to compare the temporal variations between SEED and NOSEED areas (the color-filled and black contours in Fig. R2), we prefer to keep the original pattern. To address your comment, we change the colormap to better show the differences among the panels. Although you mentioned the possible issues about jet (or rainbow) colormap, we tested several different colormaps and this reveals the differences best.

[Figure]

Fig. R2. Time-height diagram of ice concentration (left panels), ice mixing ratio (middle panels), and cloud water mixing ratio (right panels) from the (a-c) Control, (d-f) EnWS, (g-i) EnAgI and (j-l) EnWS/EnAgI experiments. The color-shading applies to the SEED areas, and the black contours are for the NOSEED areas.

Line 275: I understand if you do not want to dive into riming and aggregation, but could you provide more information on that? Maybe an appendix figure? Riming and aggregation should occur especially with these high ice concentrations.

Reply: The riming and aggregation rate are rather minor compared to the diffusional growth rate for such a thin cloud. The magnitudes of riming and aggregation rates are

more than 7 orders of magnitudes lower than the diffusional growth rate, which are negligible.

Line 279: I believe there is a word missing before "became lower ..." or what do the authors refer to? The deposition rate?

Reply: We apologize for the lack of clarity in this explanation. It means that the deposition rate became lower. The relevant sentence has been modified.

Line 309: enchantment to enhancement?

Reply: We appreciate your correction. "enchantment" is changed to "enhancement".

Line 362: I might understand Figure 12 wrong, but enhanced turbulence does not enhance both cloud glaciation and turbulent mixing. I mostly see a reduction in the time for cloud glaciation, while for turbulent mixing this is more difficult to see. Maybe a quantification could be helpful here.

Reply: We appreciate your comment. Stronger turbulence enhances the spread of the seeding plume; therefore, when calculating the characteristic time of mixing, the EnWS and EnAgI/EnWS experiments have larger $L$ (in Eq. 1) than the Control and EnAgI experiments at any moment. The original sentence caused a misunderstanding, it is now changed to *"By comparing the right and left panels in Fig.12, it can be seen that the enhanced turbulence accelerated the cloud glaciation, and the characteristic time of mixing became larger due to the enhanced spread of seeding plume"*.

Figure 13: How can you have negative condensation rates (evaporation) with vertical velocities larger than w*? Can you quantify how often you encounter which conditions, i.e., both growing or only ice crystals growing?

Reply: The $w*$ used in the calculation using Eq. 3 is based on constant temperature and pressure conditions (cloud top), primarily intended to demonstrate the overall distribution of condensation rates when vertical velocities are above or below $w*$. However, temperature and pressure within clouds are not constant. In addition, in the model, the simulation of condensation rate does not exactly follow Eq. 3, there are other factors, such as ice shape, that also influence the result. Therefore, there could be a low occurrence of negative condensation rates even when vertical velocities are slightly

above $w^*$. For the Control and EnAgI/EnWS experiments at 03:30 MT, the occurrence of simultaneous liquid and ice growth is 0.21 and 0.04, the occurrence of ice growth only is 0.79 and 0.91, and the occurrence of simultaneous liquid evaporation and ice sublimation is 0 and 0.05, respectively.

Line 397-401: this is one sentence, can you split it up?

Reply: We appreciate your comment. The original sentence has been divided into two separate sentences. *"The case presented here is a well-capped, shallow (~500 m deep) decoupled stratus cloud with a cloud top of -16°C. The results show that stronger shear-driven turbulence can enhance the dispersion of AgI particles and the nucleation and growth of ice crystals."*

Line 414: There are other (older) reference to ice crystal growth across temperatures, please add them.

Reply: The following references are added:

Chen, J.-P. and Lamb, D.: The theoretical basis for the parameterization of ice crystal habits: Growth by vapor deposition, J. Atmos. Sci., 51, 1206–1222, https://doi.org/10.1175/15200469(1994)051<1206:TTBFTP>2.0.CO;2, 1994.

Fukuta, N. and Takahashi, T. The growth of atmospheric ice crystals: A summary of findings in vertical supercooled cloud tunnel studies, J. Atmos. Sci., 56, 1963–1979, https://doi.org/10.1175/15200469(1999)056<1963:TGOAIC>2.0.CO;2, 1999.

Harrington, J. Y., Moyle, A., Hanson, L. E., and Morrison, H.: On Calculating Deposition Coefficients and Aspect-Ratio Evolution in Approximate Models of Ice Crystal Vapor Growth, J. Atmos. Sci., 76, 1609–1625, https://doi.org/10.1175/JAS-D-18-0319.1, 2019.

Line 417: Nimbostratus produce natural precipitation. Why did you choose this cloud type as an example? Is it really ideal to be seeded?

Reply: Deep stratiform clouds, such as nimbostratus, may be suitable for seeding, but it is difficult to detect unambiguous seeding signatures due to natural precipitation. There is supercooled liquid water in these clouds, and some previous studies tried to investigate the seeding effect in these clouds (e.g., Pokharel et al., 2015, Jing et al.,

2015). Although observing the seeding effect is different in these clouds, we can still use model simulations to study the impacts of turbulence.

Line 448: What ways do you see to parameterize turbulence in connection to cloud microphysics? Are there open questions in this regards?

Reply: Numerical weather predicting and climate models cannot resolve turbulence, so it is necessary to parameterize the impacts of turbulence on cloud microphysics, but it is challenging. Several studies have tried to develop parameterizations of turbulence-induced droplet collision (e.g., Franklin, 2008), but to our knowledge, there is no parameterization of ice growth affected by turbulence. Recently, we developed an observationally constrained parameterization of liquid-ice mixing inhomogeneity in ice growth for climate model, which is relevant to turbulence, but it has not been published (Yang et al., GRL, in revision).

Franklin, C., A warm rain microphysics parameterization that includes the effect of turbulence. J. Atmos. Sci. 2008, 65, 1795–1816.

Yang, J., et al., Parameterizing the heterogeneous liquid-ice mixing in modelling ice growth through the Wegener-Bergeron-Findeisen process in CAM6, Geophys. Res. Lett. In revision.

---

## Author Comment (AC2)

The reviewer's comments are in black, and responses are in blue.

Chen et al. investigates how turbulence modulates the effects of AgI cloud seeding in shallow stratiform clouds over flat terrain. Using LES simulations based on a real seeding event over North China on 20 January 2022, this study creates an artificially enhanced shear layer and finds that increased turbulence enhances AgI dispersion, ice nucleation, and crystal growth, which accelerates faster glaciation. Although stronger turbulence fosters enhanced SLW generation, the rate of ice growth outpaces condensation leading to a rapid depletion of SLW and faster transition to glaciated cloud. Turbulence amplifies short term precipitation directly downwind of seeding, it suppresses further downwind precipitation. This study highlights the delicate balance between turbulent mixing and glaciation in determining the spatial extend and efficiency of cloud seeding impacts. Overall, I felt like the study was well written but could benefit from several clarifications to enhance the study. Therefore, I am recommending major revisions.

Reply: We appreciate your insightful comments. The paper has been revised accordingly and has been improved. Please see our point-by-point response below.

General Comments:

In Fig. 1, the visible and infrared imagery show a clear depression in the cloud top along the flight track. However, it would be beneficial to demonstrate whether this displacement is consistent with the large-scale flow pattern. Specifically, could the flight track be transposed or advected downwind using the wind vectors from the ERA5 reanalysis fields shown in Fig. 1a–b? This would better contextualize the observed cloud top depression and support the attribution of the cloud signature to seeding effects.

Reply: We appreciate your comment. The wind at the seeding height is southwest, as indicated by the wind barb in the pink box in Fig. 1a. Consequently, the seeding trajectory in Fig. 1c presents northeastward displacement over time. The wind speed was about 10 m/s, and the observed depression was about 20 km downwind of the seeding trajectory. Since the observation was about 30 minutes after the seeding, indicating an advection distance of 18 km, it can be concluded that the observed displacement is consistent with the large-scale flow pattern, which facilitates the attribution of cloud signatures to seeding effects.

Similarly, in Fig. 2, the location of the S-band radar relative to the flight track is unclear and should be explicitly marked on a map. The reflectivity evolution shows a clear enhancement along one of the flight legs but not the other—why does the enhanced reflectivity from the eastern leg fade later in time? Was there a difference in cloud properties, wind shear, or moisture that could have contributed to the disparity between the seeded legs?

Reply: We appreciate your comment. The ground-based S-band radar is located north of the flight area. This is added to Fig. 1. The seeding started from the eastern flight leg, resulting in an initial enhancement of radar reflectivity on the east side. Subsequently, this reflectivity gradually diminished as precipitation particles fell. The delayed enhancement of radar reflectivity on the western side is due to the later seeding in the flight leg, while the differences in cloud properties, wind shear, or moisture may also affect the reflectivity intensity of different flight legs, but this cannot be revealed using idealized LES model.

The choice of background aerosol concentration ($N_0 = 4000$ cm$^{-3}$) seems quite high for a wintertime stratiform cloud and may not be representative of the conditions aloft over rural North China. Is there observational support for this aerosol profile? Were vertical variations in aerosol concentration considered, and how sensitive are the results to these assumptions? Given that aerosol background can strongly influence microphysical pathways, this deserves more justification or discussion.

Reply: According to previous studies, the aerosol number concentration over the North China Plain in winter typically ranges between $10^3$ - $10^4$ cm$^{-3}$ (Zhang et al., 2020; Wang et al., 2024). Sorry, we do not have measurements of aerosol concentration for this case. To test the impacts of aerosol concentration on the results, we designed an experiment considering enhanced wind shear, seeding, and natural ice nucleation with a background aerosol concentration of 2000 cm$^{-3}$. The results show that the modelled radar reflectivity and changes in precipitation induced by seeding have similar magnitudes compared to the case with higher concentrations. The reason is that in this case there is no warm rain process, and the ice particles grew through the vapor diffusion and WBF process, which are mainly controlled by the liquid water content rather than the droplet concentration. For other cases in which the microphysics are

sensitive to the droplet concentration, the seeding effect would be different between cases with clean and polluted environments. This is out of the focus of our paper, but it would be interesting to investigate the aerosol impact in the future, this discussion is added in the manuscript in Section 2.2.

[Figure]

Figure R1. East-west cross-sections of reflectivity from the experiment with (a) $N_0 = 4000$ cm$^{-3}$, and (b) $N_0 = 2000$ cm$^{-3}$ at 03:00 Model Time. Natural ice nucleation and enhanced wind shear are allowed in this case. The cross-sections are selected at y = 20 km.

[Figure]

Figure R2. Maps of difference in cumulative precipitation compared to the average natural precipitation from the experiments with enhanced shear for (a) $N_0 = 4000$ cm$^{-3}$, and (b) $N_0 = 2000$ cm$^{-3}$.

A brief explanation of how cloud was forced and maintained in the model would benefit the reader. While the use of a constant profile and a single sounding is mentioned (Section 2.2), it was not immediately clear how the cloud was maintained within the LES framework, particularly given the absence of evolving large-scale forcing. I found myself needing to read through this section twice to piece together how the cloud

structure was sustained. A concise summary early in the model setup section—perhaps explicitly stating the role of the initial liquid water mixing ratio, the static thermodynamic profile, and discussing how cloud was maintained after it was initialized— would improve clarity.

Reply: We appreciate your comment and apologize for the lack of clarity in this explanation. Since this is an idealized LES driven by a single real sounding, it is necessary to artificially introduce the cloud water mixing ratio at the initial time. The cloud layer is located at the height where the water vapor is saturated, and an inversion layer exists above the cloud top. An adiabatic cloud water mixing ratio is used. This allows supercooled liquid water to persist below the inversion, enabling the cloud to be maintained for a long time in the model. This explanation is added to the paper.

The LES is driven by a single sounding with no horizontal heterogeneity or time-evolving large scale forcing. This idealized setup limits realism, especially in terms of representing synoptic evolution and mesoscale variability. How might the results differ if the full 3D ERA5 meteorological fields were used to initialize and force the model? Would the cloud form in a similar vertical structure, and would radar reflectivity fields appear more similar to observations? A brief discussion of this limitation would improve transparency.

Reply: We appreciate your comment. With accurate 3D reanalysis data used to drive the model, the model results may be more consistent with observation. The reason we used idealized LES is that we focused on the impact of turbulence on the seeding effect. We hope the dynamics that influence liquid water formation and cloud glaciation are only driven by turbulence, so the conclusions would not be contaminated by any influence from larger-scale dynamics. If turbulence is imposed on a larger-scale dynamic forcing, it may have different impacts on cloud microphysics such as enhancing the riming and aggregation processes, which may in turn result in a different impact on the seeding effect.

One of the more confusing aspects of the paper is the comparison setup (especially compared to other seeding studies). The so-called "Control" simulation includes AgI seeding, but there is no true baseline simulation without any seeding. This makes it difficult to isolate the seeding effect, especially since fall streaks and reflectivity

enhancements appear in the Control run, which undermines the attribution of microphysical or dynamical responses to seeding. Including a no seeding case—or clearly differentiating it from the "Control"—would clarify the results and interpretation. I think running a true control and comparing it to the observations and other simulations would add a lot of value to the manuscript.

Reply: We appreciate your comment and apologize for the lack of clarity in the comparison setup. In this study, the field measurement had seeding performed, and we primarily focused on the impact of turbulence on the cloud seeding effect, so we didn't consider the simulation without any seeding. Since the seeding signature is unambiguous, analysis between SEED and NOSEED areas inherently provides the necessary contrast between seeding and no seeding simulations. To address your comments, we have renamed all the experiments. The revised names will facilitate the comparison between different experiments and clarify the results. The modified table of the experiment design is presented below, with related revisions added throughout the paper:

Table 1. Design of numerical experiments.

| Experiments | Natural ice nucleation | Enhanced wind shear | Enhanced AgI concentration |
|---|---|---|---|
| Control | No | No | No |
| NI | Yes | No | No |
| NI_WS | Yes | Yes | No |
| NI_AgI | Yes | No | Yes |
| NI_AgI_WS | Yes | Yes | Yes |

The simulations use only a one-hour spin-up period before seeding is introduced at 2:00 MT. However, at 2:30 MT (the first output shown in Fig. 7), the cloud field and reflectivity structure still appear immature. For instance, in all simulations including the Control, reflectivity increases dramatically in later time steps. This raises the possibility that early reflectivity enhancements are a spin-up artifact rather than a response to seeding. A longer spin-up or justification for the current choice is needed,

particularly since the cloud system being simulated is shallow and sensitive to small thermodynamic changes. Also, in Fig. 7, the radar reflectivity differences between SEED and NOSEED areas are subtle, especially in the early time steps. The seeding signature is difficult to identify without annotations or overlays. Annotating the plots to highlight the SEED plume location, expected signal from seeding, and NOSEED comparison regions would make the figures much more interpretable. As it stands, the reader must infer these spatial relationships without guidance.

Reply: We appreciate your comment and apologize for the misunderstanding caused by this description. The statement that "a spin-up time of at least 1 hour is needed" is because noticeable differences between the default and enhanced wind shear experiments in Fig. 7 begin to appear after 1 hour. In fact, we implemented a two-hour spin-up period to ensure the sufficient development of the cloud. This is now clarified in the revised paper. To highlight the SEED areas, we added two arrows pointing to the plumes in Fig. 7a.

While the discussion offers some thoughts on generalizability, the study's findings are drawn from a highly specific case: a shallow, capped stratiform cloud in a quiescent wintertime environment. The conclusions regarding turbulence-enhanced glaciation and the transition from positive to negative seeding effects may not extend to deeper, more dynamic clouds. A more tempered framing of the conclusions, emphasizing the case-specific nature of the results, would strengthen the overall presentation.

Reply: We appreciate your comment. The conclusions have been revised accordingly, with particular emphasis on the case-specific limitations of the results.

*"For shallow, capped stratiform clouds in a quiescent wintertime environment, stronger turbulence can accelerate the seeding effect by enhancing AgI particle dispersion, ice nucleation, and ice growth through the WBF process, resulting in faster consumption of LWC and cloud glaciation in the SEED area, even though stronger turbulence also enhances the liquid water formation."*

*"It is the competition among liquid condensation, mixing, and cloud glaciation that determines the downwind effect of glaciogenic cloud seeding. For the shallow cloud presented in this paper, neither the liquid condensation nor the turbulent mixing can overcome the cloud glaciation intensification by turbulence."*

*"This study is based on case analysis with limitations. To further understand the role*

*of turbulence in natural and seeded clouds under different conditions, more*
*observational and modelling studies are needed in the future."*

Specific Comments:

Line 103 – "...brightness temperature increased by about 2 °C..." Is this value within the noise level of the satellite product? What's the uncertainty?

Reply: The uncertainty of brightness temperature measurement is within 1 °C (Geng et al., 2020).

*Geng, X., Min, J., Yang, C., et al. 2020. Analysis of FY-4A AGRI Radiance Data Bias Characteristics and a Correction Experiment. Chinese Journal of Atmospheric Sciences (in Chinese), 44(4): 679−694. doi:10.3878/j.issn.1006-9895.1907.18254*

Line 116 – "...radar echo appeared about ten minutes after seeding..." Is this consistent with modeled ice nucleation onset time? If not, why not? Feel like this was not discussed later in the text.

Reply: In observation, it takes about 10 minutes for the ice to grow big enough to be detected by radar. In the model, ice nucleation started at 02:00 MT (Model Time), and it also took about 10 ten minutes to see the radar signature (as shown in Fig. R3 below). So the model is consistent with observation.

[Figure]

Figure R3. Map of composite reflectivity from Control experiment at 02:10 Model Time.

Line 135: "...periodic lateral boundaries..." Could periodic boundaries introduce

artifacts in this stratiform cloud case?

Reply: This study employs an idealized LES to simulate stratiform clouds, which do not consider large-scale forcing effects. The physical quantities within the model domain maintain horizontally homogeneous distributions. Under this condition, the periodic boundary will replicate this homogeneity without introducing artifacts.

Line 158 – "A single sounding is used to drive the model…" Can the authors clarify when and where the sounding was launched relative to the seeding flight track and time? Was it an operational sounding, and how representative is it of the cloud environment during the seeding event? Spatial and temporal offsets could influence the initial conditions and evolution of the simulated cloud.

Reply: The sounding was launched at the Luancheng station (which is located in the research area shown in Fig. 1) at 00:00 UTC on 20 Jan 2022. We added the relative humidity to the profiles. It is seen that the sounding well captures the saturated cloud layer and the inversion above it. Since the station is in the research area, the impact of spatial offsets would be small.

[Figure]

Figure R4. The initial vertical profiles of (a) temperature and potential temperature, (b) actual vapor mixing ratio, saturation vapor mixing ratio relative to water and relative humidity, and (c) original and enhanced U and V components. The grey shaded area in (b) indicates the initial liquid water mixing ratio.

Line 164: "...Richardson number...from 16.81 to 0.67..." This is a large reduction. Can you show whether turbulence actually became dynamically unstable (e.g., Ri < 0.25)?

Reply: The bulk Richardson number was never smaller than 0.25 in this study. In

general, Ri < 0.25 indicates dynamic instability, but turbulence can also be triggered for 0.25 < Ri < 1 (e.g., Galperin et al., 2007), Ri > 1 indicates laminar flow. In this study, Ri was decreased to 1/25 of its original value (from 16.81 to 0.67) at the height of 1519–1733 m, when the wind shear was enhanced by five times, indicating favorable conditions for turbulence development, but not enough to achieve dynamic instability. *Galperin, B. et al., On the critical Richardson number in stably stratified turbulence. Atmos. Sci. Lett., 2007, 8, 65-69.*

Line 265: "...IWC up to 0.06 g/kg..." How does this compare with in-situ observations from similar seeded clouds?

Reply: We apologize that there are no in-situ measurements of IWC for this case. However, the magnitudes of radar reflectivity are similar between the model and observation, and there is a positive relationship between IWC and radar reflectivity based on many previous studies.

Line 286: "...generation of liquid water was significantly slower..." Can this be quantified more directly using a rate ratio (e.g., production/consumption)?

Reply: In the model, the liquid water condensation rate is calculated based on mass change per unit time. A negative condensation rate indicates that the consumption of liquid water exceeds production. The statement that "the generation of liquid water was significantly slower than its consumption by ice growth" refers to the condition with weaker turbulence. As shown in Fig. 9c, enhanced turbulence results in a notably decreased condensation rate, indicating an accelerated consumption rate.

Line 311: "...cumulative precipitation decreased after 4:40 MT..." What was the baseline precipitation without seeding during this period?

Reply: Under default wind shear condition, the cumulative precipitation in the NOSEED area was $3.61\times10^{-3}$ mm, $4.04\times10^{-3}$ mm, and $4.48\times10^{-3}$ mm at 04:40 MT, 04:50 MT, and 05:00 MT. With default AgI amount and wind shear, the differences in cumulative precipitation between SEED and NOSEED areas remained small after 04:40 MT.

---

## Author Comment (AC3)

The reviewer's comments are in black, and responses are in blue.

Glaciogenic cloud seeding is studied through numerical simulations of a case study of a shallow, stratiform cloud comprised of supercooled liquid water. WRF LES is employed for these simulations with a fast spectral bin microphysics scheme. Four simulations are undertaken to test the relative importance of turbulence generated by vertical winds shear (control vs enhanced) and the concentration of the released AgI seeding agent (control vs enhanced).

The primary conclusion highlights the importance of enhanced turbulence in accelerating glaciogenic seeding in comparison to enhanced AgI concentrations.

The paper is, in general, well-written with appropriate figures laid out in the logical manner. The literature review/introduction was particularly well done. I did find the writing to be fairly repetitive, especially with respect to the effect of turbulence through the results, enough so that I am specifically commenting about it. At other points, I thought some material was missing, as detailed below.

I respect and appreciate that the author readily acknowledges the inherit sensitivity of weather modification to the specific meteorology/event, what works in this case study may not work for a different event.

The majority of my comments pertain to further clarification of the work and assumptions rather than making new simulations. As such I consider the comments to be minor in nature.

Reply: We appreciate your insightful comments. The paper has been revised accordingly and has been improved. Please see our point-by-point response below.

Why do you want to run in two different times, with the case study being in UTC and the simulations being in model time? This made it more difficult to go back and forth between the two. Does seeding at 2:00 MT correspond to ~ 8:00 UTC? Does this mean the simulations are spun up for two hours, rather than one? Given that 8:45 UTC is 16:45 CST, would the local time be around local dusk for January, which is why the visible image has no colour?

Reply: Yes, the simulations are in model time. Unlike realistic LES, the idealized LES does not follow actual world time but rather represents the internal temporal framework of the model. The input is a single sounding measurement rather than real 3D data such

as reanalyses. Therefore, the modeled time cannot be synchronized with field observations. Yes, the model spun up for two hours before seeding, this is now clarified in the revised paper.

Figure 1: Given that the simulations only go up to 3 km, I'd rather see the MSLP and surface temperature rather than the 500 hPa geopotential height. In particular, I'd like to know more about the boundary layer stability and boundary layer dynamics, beyond "quiescent and stably stratified". Do we have warm air advection or cold air advection? The satellite imagery suggests that this solid SLW cloud layer is moving up from the south, as discussed in the manuscript.

Reply: We appreciate your comment. Fig.1b is modified to show the MSLP, surface temperature, 10-m wind and BV frequency of cloud layer (Fig. R1b). Yes, there was warm air advection as suggested by the wind veering. The cloud layer BV frequency was positive, indicating stably stratified environment. This also can be seen from the sounding measurement which shows the boundary layer height was low, and the cloud layer was stable.

[Figure]

Figure 1. (a) Synoptic conditions at 850 hPa in North China at 06:00 UTC on 20 Jan 2022 obtained from ERA5 reanalysis data, including the geopotential height (dam, blue contours), isotherms (°C, red contours), wind barbs, and relative humidity (shaded).

The yellow star indicates the location of radar and the magenta box is the flight area. (b) Map of sea-level pressure, surface temperature, 10-m wind and BV frequency squared at cloud layer (1.3-1.8 km above ground level). (c) Visible image and (d) brightness temperature at 12 μm obtained from FY4A satellite at 08:45 UTC. The red lines indicate the seeding trajectory.

Figure 3: When and where was the sounding taken? Was this from the seeding aircraft or an upper air sounding?

Reply: It was an upper air sounding, launched at the Luancheng station (which is located in the research area shown in Fig. 1) at 00:00 UTC on 20 Jan 2022.

I am confused between Figure 5, which has a steady cloud comprised of SLW, and Figure 7, which has ice going to the surface and Figure 8, which has the cloud water mixing ratio diminishing through the simulation. That is directly at odds with Figure 5. I assume that this is a consequence of this being 'the control simulation without natural ice simulation' while Figures 7 & 8 allow natural ice nucleation. This needs to be explained in much greater detail. Would a true 'control' run (i.e., no enhanced shear and no seeding at all) glaciate just like the NOSEED areas in Figure 7? Is the reflectivity in Figure 7 being caused by ice failing below the cloud deck? I assume this does not match reality. What does this suggest about the formation of ice in the simulations? It would be best not to gloss over this.

Reply: We appreciate your comment. In the revised paper, we changed the names of the experiments to avoid such confusion (Table 1). Now, the "control" run is exactly the same as the field measurement: no natural ice nucleation, no enhanced wind shear, but seeding is performed. The comparison between the control run and observation (Fig. 5 in the paper) indicates the model can well capture the characteristics of the seeding effect. The radar reflectivity outside of the seeding plume in Fig. 7 is the result of natural ice nucleation and the changes in radar reflectivity indicate the falling snow (below the cloud base). We turned on natural ice nucleation because other than the seeding effect, we also want to investigate how enhanced turbulence would affect the natural ice microphysics and precipitation (outside of the seeding plume), and we see enhanced ice growth and precipitation by stronger turbulence (Fig. 7 and 8).

Table 1. Design of numerical experiments.

| Experiments | Natural ice nucleation | Enhanced wind shear | Enhanced AgI concentration |
|---|---|---|---|
| Control | No | No | No |
| NI | Yes | No | No |
| NI_WS | Yes | Yes | No |
| NI_AgI | Yes | No | Yes |
| NI_AgI_WS | Yes | Yes | Yes |

Line 245: magenta lines? The magenta line is at the inversion and is clearly visible at 4:00 MT.

Reply: Sorry for the misunderstanding. Here, we mean the magenta lines (liquid layer top) disappeared above the seeding plume at 4:00 MT, indicating the supercooled liquid water was consumed by the seeding. In areas unaffected by seeding, the liquid layer persisted and the magenta lines are still present. This is now clarified in the revised paper.

I am 90% confident that SEED area refers to the two primary red streaks in Figure 7 and NOSEED refers to everything else. I think, however, a couple of arrows pointing to what you are calling SEED would be very helpful.

Reply: We appreciate your comment. A couple of arrows are added to point to the SEED plumes in Fig. 7 in the revised paper.

Given that there is no source of water (latent heat flux) in the simulations, it seems obvious to me that any enhancement of precipitation upwind will require less precipitation at some point downwind. (I.e., robbing Peter to pay Paul.) This is just a statement of the conservation of water. If the simulation was run longer and the domain was bigger, I would expect to see the same thing for the non-shear cases. Am I mistaken? Ultimately, I don't think you can really address this questions using LES simulations with periodic boundary conditions. Once the precipitation reaches the surface, it's gone.

Reply: We appreciate your comment. In fact, within the model domain, we do not see a "robbing Peter to pay Paul" effect in the non-shear case. Although the ice concentration is enhanced by seeding, the formation of supercooled liquid water was still faster than its consumption by ice growth in the downwind areas. We agree that it is possible that the precipitation may decline somewhere out of the model domain, which may require a much larger domain in a real-case simulation. This is out of the

focus of this study, which primarily shows turbulence can accelerate the seeding effect. It would be interesting to study the downwind effect in a larger domain in real cases in the future.

---

## Author Response (AR2)

The reviewer's comments are in black, and responses are in blue.

Thank you to the authors for thoroughly answering the comments of all reviewers. I only have minor comments in this second round, which can be swiftly addressed before acceptance of the manuscripts.

- The authors nicely responded to the reviewer comments, but many points / questions raised have not been implemented in the manuscript. I would like to see more of the resolved comments also in the main manuscript, as also other readers may stumble upon the same questions and should not be forced to check the open discussion documents. This goes for the SEED / NOSEED definition, ice nucleation, precipitation significance (which actually was not properly discussed after reviewer comments, even though indicated), signatures of the observed radar reflectivites, etc.

- Colormap Jet: I have to disagree with the authors, because there are various colormaps allowing for more contrast among a broader range of values. I am adding a few examples here: PyART is a specific package for colormaps for remote sensing instruments and the scientific colormaps by Fabio Crameri have been designed specifically for the purpose of allowing maximum distinguishability while having color-vision deficiency friendly colormap. I am aware that jet is a prominent colormap, especially in the remote sensing community, however, it has been proved to induce biases and not being color vision deficiency friendly, and with that contradicts the guidelines by ACP. I strongly urge the authors to adapt their figures. Moreover, adequate ranges in the colormap could help to increase the contrast in the signals.

Upon these changes (which should be minor in nature), I am fully recommending the manuscript for publication.

Reply: We appreciate your comments and the paper is revised accordingly. The definitions of the SEED and NOSEED areas have been revised in Line 203 of the manuscript, and an explanation of why no-seeding simulations were not designed has been added in Line 210. The description of natural ice nucleation in the model has been added in Line 155, while the discussion on whether to activate the natural ice nucleation is presented in Line 206. The discussion of precipitation significance (i.e., simulation

uncertainty) is presented in Line 367, while the calculation methodology for differences in SEED and NOSEED areas and the clarifications of the numbers in the text have been added in Lines 336 and 368, respectively. Signatures of the observed radar reflectivity have been supplemented in Line 126. Additionally, some omitted explanations have been added into the manuscript, including new clarifications in Lines 266-268 that resolve potential ambiguities in Fig. 7. The figures related to the jet colormap have been recreated as follows, which are now match the guidelines by ACP:

[Figure]

Figure R1. (a) Synoptic conditions at 850 hPa in North China at 06:00 UTC on 20 Jan 2022 obtained from ERA5 reanalysis data, including the geopotential height (dam, blue contours), isotherms (C, red contours), wind barbs, and relative humidity (shaded). The yellow star indicates the location of radar and the magenta box is the flight area. (b) Map of sea-level pressure, surface temperature, 10-m wind and BV frequency squared at cloud layer (1.3-1.9 km above ground level). (c) Visible image and (d) brightness temperature at 12 microns obtained from FY4A satellite at 08:45 UTC. The blue lines

indicate the seeding trajectory.

[Figure]

Figure R2. The radar reflectivity at 1.5° elevation from 08:06 UTC to 08:48 UTC measured by a ground-based S-band radar located north of the flight area.

[Figure]

Figure R3. (a-d) Maps of composite reflectivity and cloud water mixing ratio from the Control experiment at seeding height from 02:30 to 04:00 Model Time. (e-h) Cross-sections of reflectivity and cloud water mixing ratios from the Control experiment along y = 40 km from 02:30 to 04:00 Model Time.

[Figure]

Figure R4. Maps of cloud top temperature in the Control experiment from 03:00 to 04:00 Model Time.

[Figure]

Figure R5. East-west cross-sections of reflectivity from (a-d) NI, (e-h) NI_WS, (i-l) NI_AgI, and (m-p) NI_AgI_WS experiments from 02:30 to 04:00 Model Time. The cyan lines indicate the top of the liquid layer. Natural ice nucleation is allowed in the simulations. The cross-sections are selected at y = 20 km, y = 20 km, y = 40 km, and y = 50 km at 02:30, 03:00, 03:30, and 04:00 MT, respectively.

[Figure]

Figure R6. Time-height diagrams of ice concentration (left panels), ice mixing ratio (middle panels), and cloud water mixing ratio (right panels) from the (a-c) NI, (d-f) NI_WS, (g-i) NI_AgI and (j-l) NI_AgI_WS experiments. The color-shading applies to the SEED areas, and the white contours are for the NOSEED areas.